# Clusters of deep intronic RbFox motifs embedded in large assembly of splicing regulators sequences regulate alternative splicing

Francesco Tomassoni-Ardori[1], Mary Ellen Palko[1], Melissa Galloux[2], Lino Tessarollo[1]*

1 Neural Development Section, Mouse Cancer Genetics Program, Center for Cancer Research, National Cancer Institute, National Institutes of Health, Frederick, Maryland, United States of America,
2 Independent bioinformatician, Marseille, France

* tessarol@mail.nih.gov

## Abstract

The RbFox RNA binding proteins regulate alternative splicing of genes governing mammalian development and organ function. They bind to the RNA sequence (U) GCAUG with high affinity but also non-canonical secondary motifs in a concentration dependent manner. However, the hierarchical requirement of RbFox motifs, which are widespread in the genome, is still unclear. Here we show that deep intronic, tightly clustered RbFox1 motifs cooperate and are important regulators of alternative exons splicing. Bioinformatic analysis revealed that (U)GCAUG-clusters are widely present in both mouse and human genes and are embedded in sequences binding the large assembly of splicing regulators (LASR). Integrative data analysis from eCLIP and RNAseq experiments showed a global increase in RNA isoform modulation of genes with Rbfox1 eCLIP-peaks associated with these clusters. Experimentally, by employing recombineering mutagenesis in a bacterial artificial chromosome containing the NTrk2 mouse region subjected to alternative splicing we showed that tightly clustered (U)GCAUG motifs in the middle of 50 Kb introns are necessary for RbFox1 regulation of NTrk2 gene isoforms expression. Moreover, our data raise the possibility that clustered (U)GCAUG-motifs promote the recruitment of RbFox proteins to form a Rbfox/LASR complex required for splicing. Altogether, these data suggest that clustered, distal intronic Rbfox-binding motifs embedded in LASR binding sequences are important determinants of RbFox1 function in the mammalian genome and provide a target for identification of pathogenic mutations.

## Author summary

In the genome there are thousands of sequences recognized by the RNA binding protein RbFox1. However, not all of them bind RbFox1 and regulate alternative

purpose. The work is made available under the Creative Commons CC0 public domain dedication.

**Data availability statement:** The eCLIP data generated for this study have been deposited in NCBI's Gene Expression Omnibus and is accessible through GEO Series accession number GSE263173. The RNAseq data is accessible through GEO Series accession number GSE263172.

**Funding:** This research was supported by the Intramural Research Program of the National Institutes of Health (NIH). The contributions of the NIH authors were made as part of their official duties as NIH federal employees, are in compliance with agency policy requirements, and are considered Works of the United States Government. However, the findings and conclusions presented in this paper are those of the authors and do not necessarily reflect the views of the NIH or the U.S. Department of Health and Human Services. National Institute of Health ZIA BC 010390 to FTA, MEP, MG, LT. The funders had no role in study design, data collection and analysis, decision to publish, or preparation of the manuscript.

**Competing interests:** The authors have declared that no competing interests exist.

splicing. We have found that clusters of 4 or more, closely associated RbFox1 binding sites embedded in sequences binding the large assembly of splicing regulators are important determinants of RbFox1 function. Bioinformatic and experimental evidence show that these clusters are mainly present in intronic regions and can regulate splicing of exons located tens of kilobases away. The discovery of these deep intron regulatory signals is important for the study of alternative splicing mechanisms and uncovering the medical significance of deep intron mutations in disease mechanisms.

## Introduction

In eukaryotes, alternative splicing of precursor mRNAs (pre-mRNAs) is a critical process involving the removal of introns and the inclusion or skipping of specific exons. In humans, more than 90% of genes are estimated to be alternatively spliced, contributing to the generation of gene-isoforms and, consequently, genome complexity. Alternatively spliced exons are flanked by longer introns compared with those flanking constitutively spliced exons suggesting that intron length may harbor increased genetic content for regulation of alternative exon choice [1,2]. However, deciphering how the content of long introns influences splicing has been difficult because many RNA-binding proteins (RBPs) can act directly or indirectly to modulate pre-mRNA splicing processes by binding to recognition motifs depending on affinity and context, and because specific sequences may lead to mRNA looping which also affects splicing [3,4]. It has also been reported that alternatively spliced introns, which are usually long, are removed last by the spliceosome although the regulatory mechanism is poorly defined [5]. Possibly, some delay may be caused by a sequential removal of small intron chunks through a stochastic recursive splicing process that is unique to longer introns (>10 Kb), whereas shorter introns (<1.5 kb) are excised in one step as complete units [6]. Nevertheless, experimentally addressing alternative splicing involving long introns has been challenging because of the lack of suitable in vitro and in vivo systems to perform such studies. *Ntrk2* belongs to the class of genes with large introns and generating different isoforms by alternative splicing. It spans about 350 Kb of the genome and by alternative splicing generates a full-length tyrosine kinase receptor (TrkB.FL) and a truncated isoform lacking kinase activity (TrkB.T1) [7]. Importantly, it includes long introns (about 50 Kb) surrounding the alternative spliced exons and the size of these introns suggests significant coding information for precise spatio-temporal expression of the TrkB isoforms in response to the many developmental and environmental cues [8,9]. Almost nothing is known about the regulatory mechanisms of TrkB isoforms expression, but proper expression of these receptors is crucial for normal development and function of the mammalian brain as their dysregulation leads to neurodevelopmental and psychiatric disorders [7]. We have reported that the RBP RbFox1 regulates the expression levels of receptor isoforms generated by the TrkB locus through a still unknown mechanism [8]. Gene targeting experiments in mouse and analysis of Rbfox1 copy number variations

in human has shown that Rbfox1 is a critical player in neuronal excitability and, like TrkB, has been associated with multiple psychiatric disorders and complex pathologies of the central nervous system (CNS) including epilepsy, autism spectrum disorders, Alzheimer and Parkinson diseases [10–17]. In mammals, Rbfox1 belongs to a family of RNA-binding proteins (RBPs) involved in alternative splicing regulation which includes three members: Rbfox1, Rbfox2, and Rbfox3 (also known as NeuN, widely used as a neuronal marker). The expression of Rbfox1 is restricted to neurons, heart, and skeletal muscle where it participates in the regulation of pre-mRNA splicing in the nucleus by binding predominantly to intronic regions [18,19]. Indeed, CLIP experiments in cell lines and in brain have shown Rbfox1 is associated mostly with intronic sequences in the nucleus [20,21,17]. Moreover, Rbfox1-isoforms preferentially expressed in the cytoplasm can also regulate mRNA stability and expression of target transcripts by binding to their 3'UTR sequence in the cytoplasmic compartment [12]. All the Rbfox family members share an almost identical and highly conserved RNA recognition motif (RRM) capable of binding with high affinity to the penta/hexa-ribonucleotide (U)GCAUG [19, 22,23]. However, RbFox proteins can also bind to secondary motifs in vitro and in vivo enabling RbFox concentration-dependent regulation of exon inclusion in neuronal differentiation and diversification [24].

To investigate how RbFox1 regulates inclusion or exclusion of the TrkB.T1 encoding exon surrounded by 50 Kb introns, we exploited the Bacterial Artificial Chromosome (BAC) manipulation technology to generate a ~ 165 Kb *Ntrk2* minigene including the intact intron/exon genomic structure of the TrkB locus (TrkB-BAC minigene) undergoing alternative splicing [25]. By enhanced RNA Cross-Linking Immuno-Precipitation (eCLIP) experiments and BAC recombineering mutagenesis in cells containing mutant BACs we found that Rbfox1 modulates the expression of the TrkB.FL and TrkB.T1 isoforms by binding to tightly clustered RbFox binding sites located up to 25 Kb from the alternative spliced exon. Moreover, we show that these clustered penta/hexa-nucleotides (T)GCAUG are widely present in distal intronic regions of the mouse and human genome, coinciding with eCLIP peaks and positively correlating with RbFox1 regulation of expression of alternatively spliced genes. Our data suggest that clustered Rbfox motifs located in far intronic regions are a general feature of the mammalian genome and are important regulators of gene-isoform expression. In addition, the Bacterial Artificial Chromosome (BAC) manipulation technology provides a new tool to study mechanisms regulating large introns splicing.

## Results

### A bacterial artificial chromosome (BAC) system to study deep intronic regions mediating Rbfox1 alternative splicing function

We have previously reported that RbFox1 regulates the expression of the receptor isoforms encoded by the *Ntrk2* gene (TrkB) [8]. This gene spans about 350 Kb across the mouse genome, generates two major receptor isoforms and includes large introns of about 50 Kb in the region where alternative splicing occurs. TrkB isoforms have identical extracellular and transmembrane domain and either a short intracellular region (TrkB.T1) or a catalytic tyrosine kinase domain (TrkB.FL) (Fig 1A). RbFox1 regulation of TrkB isoform expression is not mediated by the isoforms unique 3'UTR, as reported for other genes [8,12]. Therefore, to investigate whether other intronic sequences regulate alternative splicing, using Bacterial Artificial Chromosome (BAC) manipulation technology [25,26] we modified a *Ntrk2* BAC containing the genomic region subjected to alternative splicing and including the extracellular juxtamembrane and the transmembrane coding exons, the TrkB.T1-specific exon and the first two exons encoding the tyrosine kinase domain (Fig 1A). The BAC is about 164 Kb and includes the two large 50 Kb introns surrounding the alternatively spliced TrkB.T1 exon (Fig 1A). A cDNA fragment encoding the missing part of the extracellular domain preceded by a synthetic CAG promoter [27] was fused in frame to the upstream extracellular juxtamembrane exon of the BAC; downstream, the missing tyrosine kinase domain coding cDNA was fused in frame to the second tyrosine kinase exon of the BAC (Fig 1A) [28]. These additions allowed for expression of the complete TrkB.T1 and TrkB.FL receptor isoforms (TrkB-BAC minigene). Importantly, it preserved the intact, endogenous genomic structure of a large portion of the gene where alternative splicing occurs.

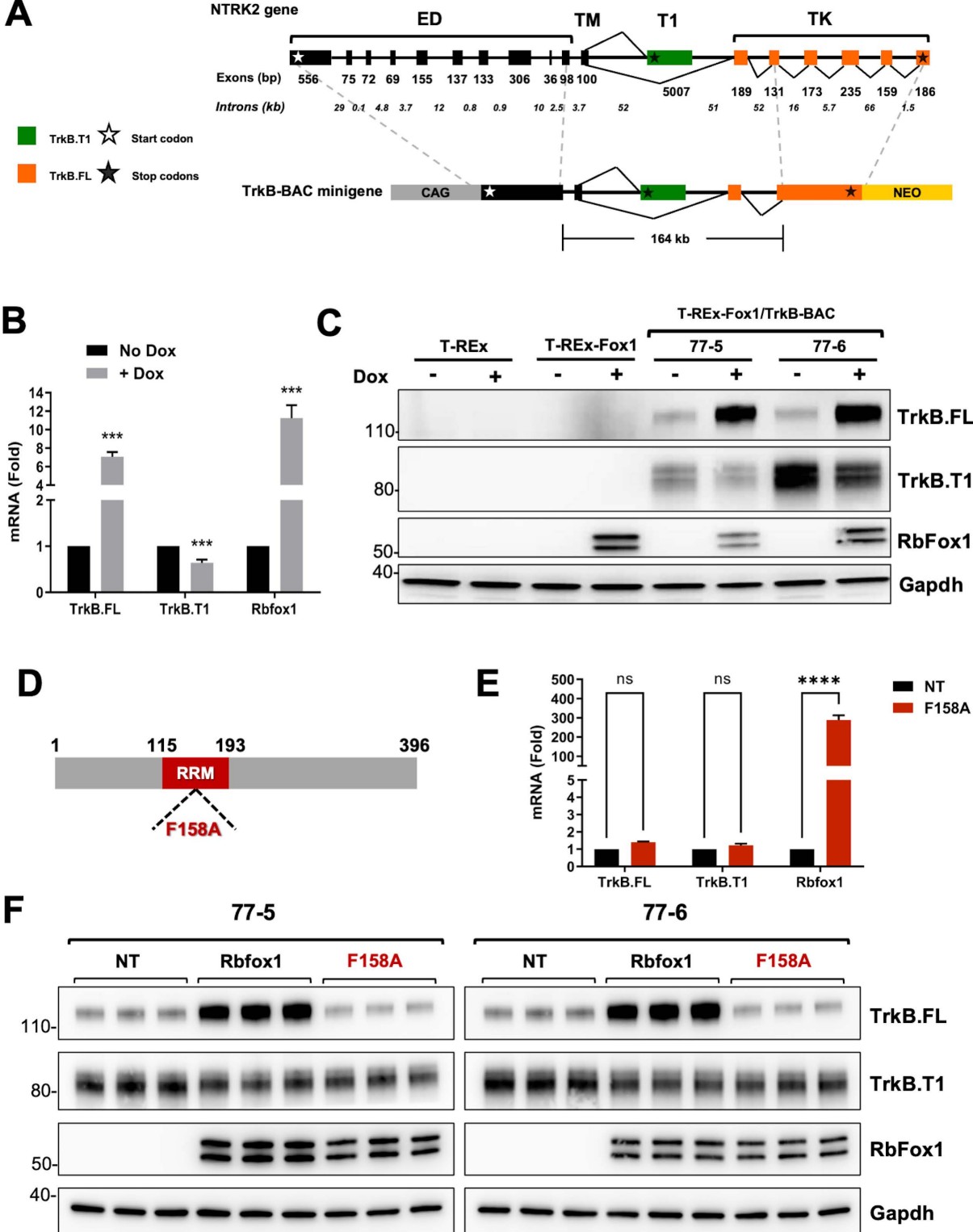

**Fig 1. Generation of a TrkB-BAC minigene system to study Rbfox1 function on TrkB isoforms expression levels.** (A) Schematic representation of the murine *NTrk2* (TrkB) gene and of the TrkB-BAC minigene. Exon length is indicated in base pairs (bp) while intron length is in kilo-bases (Kb).

White and black stars indicate the start and stop codons, respectively. Exons in black encode for the extra-cellular (ED) and the transmembrane (TM) domain and are common to both the truncated (TrkB.T1) and full length (TrkB.FL) isoforms. In green (T1) is the exon unique to the TrkB.T1 isoform. In orange are the exons specific to the TrkB.FL isoform and encoding the tyrosine kinase domain (TK). The TrkB-BAC minigene includes a murine 164 Kb genomic fragment with an upstream synthetic CAG promoter and a cDNA fragment encoding the extracellular domain fused in frame to a juxtamem-brane domain exon, and, downstream, a cDNA fragment encoding the TK domain and a neomycin resistance cassette (NEO) used for selection. (B) Quantitative PCR analysis of TrkB.FL, TrkB.T1 and Rbfox1 expression from two independent HEK293 clones expressing the TrkB-BAC minigene (77−5 and 77−6 cells) in the absence (No Dox) or presence (+ Dox) of doxycycline (0.5 µg/ml for 48h); n = 6 ± SEM (n = 3 for each clone). (C) Western blot analysis of lysates from parental T-REx 293 cells (T-REx), T-REx 293 cells with knocked-in Rbfox1 under doxycycline control (T-REx-Fox1), and the two clones expressing the TrkB-BAC minigene (77−5 and 77−6) with or without doxycycline (Dox: 0.5 µg/ml for 48h) and probed with an antibody recogniz-ing TrkB.FL or TrkB.T1 specifically. An antibody against RbFox1 was used to verify RbFox1 induction and Gapdh was used as a loading control. Note that the parental T-REx cell line does not express RbFox1 while the engineered cell line and the 77−5 and 77−6 minigene-expressing clones express RbFox1 only in the presence of doxycycline. (D) Schematic representation of the RbFox1 protein showing the location of the RNA recognition motif (RRM in red) and the position of the F158A mutation within the RRM motif. (E) Quantitative PCR analysis of TrkB.FL, TrkB.T1 and Rbfox1 expression levels from two independent clones with the TrkB-BAC minigene not transfected (NT) or transfected with the Rbfox1-F158A mutant expressing plasmid; n = 6 ± SEM (n = 3 for each clone). (F) Western blot analysis as in (C) from two independent clones with the TrkB-BAC, untransfected (NT) or transfected with a plasmid expressing wild-type Rbfox1 (Rbfox1) or the mutant Rbfox1-F158A (F158A).

Next, we introduced the TrkB-minigene into T-REx-293 cells, a human HEK293 cell line engineered to express a specific gene, in our case RbFox1, under the control of the tetracycline inducible system by Doxycycline (Dox) [8,29]. Importantly, T-REx-293 cells do not express TrkB receptors endogenously greatly facilitating the interpretation of the expression data obtained from the minigene in the absence or presence (+Dox) of Rbfox1. Two independent T-REx-293;Rb-Fox1clones with the TrkB minigene confirmed expression of both TrkB.T1 and TrkB.FL. Strikingly, in both clones, Rbfox1 expression changed the levels of the TrkB isoforms. Specifically, Rbfox1 significantly increased TrkB.FL isoform at both protein and mRNA levels while TrkB.T1 decreased, although to a lesser degree (Fig 1B). These data suggest that the TrkB-minigene contains elements mediating Rbfox1 function on TrkB isoform alternative splicing. Note that although trun-cated TrkB appears as multiple bands in western analysis this is a consequence of different TrkB.T1 glycosylation levels as confirmed by de-glycosylation experiments and RT PCR analysis (S1 Fig).

To investigate whether the RNA-binding function of Rbfox1 is required to regulate alternative splicing of the TrkB mini-gene, we tested a mutant Rbfox1 with impaired RNA-binding activity (F158A) [22,30,54]. As shown in Figs 1D–1F, trans-fection of the Rbfox1-F158A into cells with the minigene failed to modulate the expression of TrkB isoforms both at the mRNA and the protein level, suggesting that a functional RNA-binding domain is essential for Rbfox1 activity.

### Tightly clustered RbFox1 binding sequences in deep intronic regions of the TrkB minigene overlap with eCLIP peaks

The requirement of a functional RbFox1 RNA-binding domain to regulate TrkB isoform expression suggests direct binding to TrkB mRNA. Therefore, to identify RbFox1 binding sites in the TrkB pre-mRNA expressed by the minigene we performed Enhanced Cross-Linking Immuno-Precipitation (eCLIP) experiments in the presence or absence of Rbfox1 [31]. Immuno-precipitation with a RbFox1-specific monoclonal antibody and RNA sequencing identified seven statistically significant eCLIP peaks across the minigene when RbFox1 was expressed. Interestingly, all peaks were located in distal intronic regions and outside the TrkB.T1 3'UTR confirming our previous result that RbFox1 does not regulate TrkB splicing through the 3'UTR ([8]; Figs 2 and S2). Curiously, two of the seven eCLIP peaks mapped on sequences about 300-nucleotides long characterized by the presence of multiple closely clustered (T)GCATG-motifs (Fig 2B, Fig 2C). These clusters, defined as 'Cluster 1′ and 'Cluster 2′, are in the middle of the two large 50 Kb introns flanking the TrkB.T1-coding exon and display, respectively, nine (T)GCATG-motifs, for the intron upstream, and seven, for the intron downstream (Fig 2B, Fig 2C). The remaining five eCLIP peaks mapped on sequences with fewer (T)GCATG-motifs (S2 Fig).

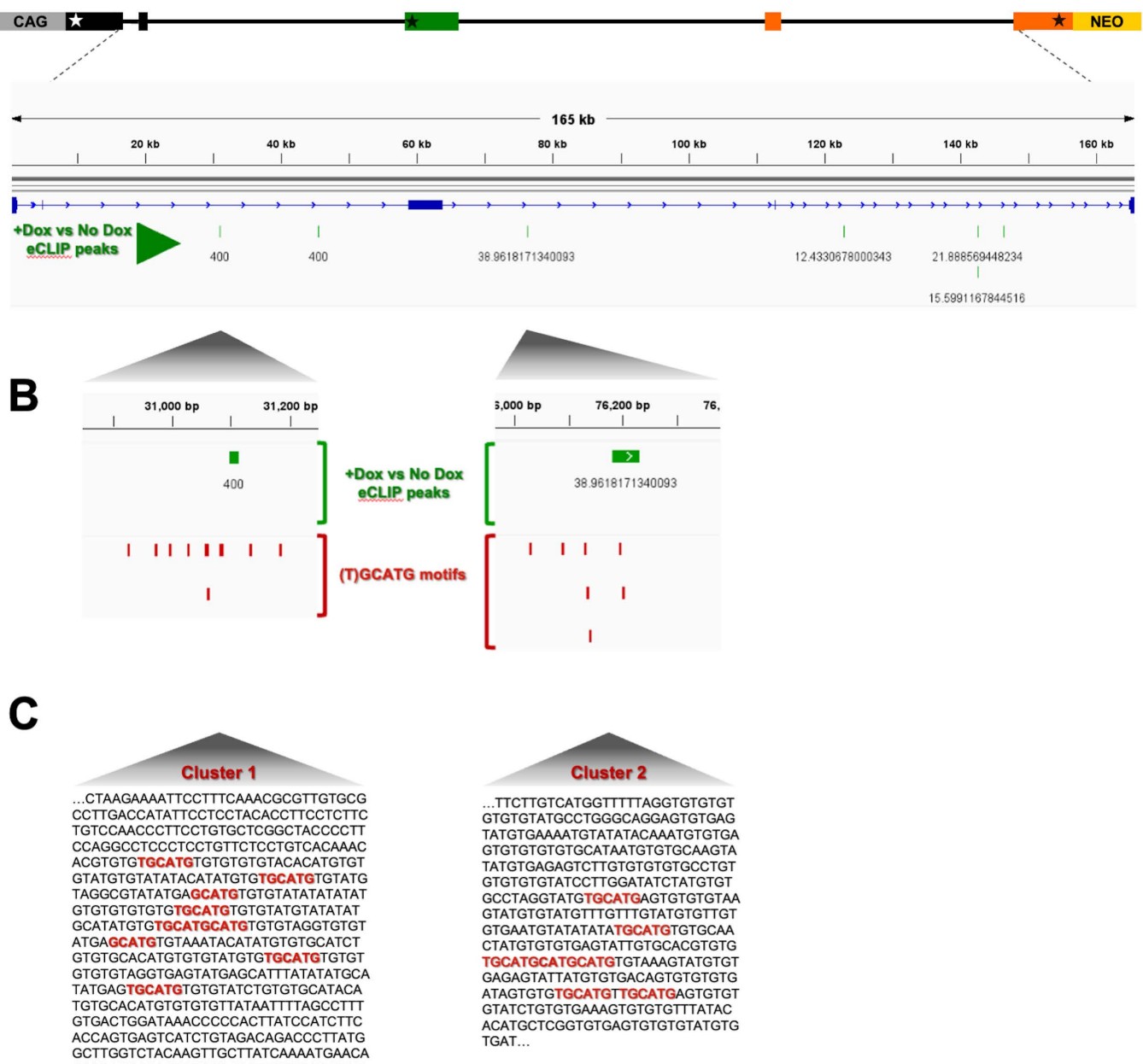

**Fig 2. Rbfox1 eCLIP peaks in a TrkB-BAC minigene coincides with clusters of (T)GCATG motifs.** (A) eCLIP analysis of RbFox1 in HEK293 cells with the TrkB-BAC. eCLIP peaks were derived by subtracting the signal obtained in the absence of RbFox1 (-Dox), considered as background, from the signal from the same cells (line 77−5 from Fig 2) expressing RbFox1. In green are seven statistically significant eCLIP peaks, all in distal intronic regions. Numbers under each eCLIP peak indicate the p-value (-log$_{10}$). (B) Enlargement of the areas containing two eCLIP peaks (green) relative to the position of RbFox1 (T)GCATG binding motifs (red). (C) Sequence of the DNA region including the two (T)GCATG-Rbfox1 motif (red) clusters shown in (B). Note that both clusters of (T)GCATG-motifs (cluster 1 and cluster 2) are located in distal intronic regions flanking the specific TrkB.T1 exon.

## Clusters of Rbfox1 binding sites are widespread in the mouse and human genome.

The finding that eCLIP peaks correspond to (T)GCATG-clusters and may impact on the regulation of TrkB isoforms expression levels prompted us to investigate whether the presence of (T)GCATG-clusters is widespread in the genome and is part of the RbFox1 regulatory function on gene expression. To screen the mouse genome, we first defined 'clusters of motifs' as genomic sites with at least four or more (T)GCATG elements confined within 500 nucleotides (Fig 3A). Overlapping clusters were considered as one, large, extended cluster.

Bioinformatic analysis revealed that the mouse genome has about twenty-one thousand (T)GCATG-clusters present in 8,348 genes and most of the clusters are located in intronic regions (Fig 3B, Fig 3C) (S1 Table). Analysis of Rb-Fox1 High-Throughput Sequencing of RNA isolated by Crosslinking Immunoprecipitation (HITS-CLIP) data from the mouse brain showed that 8356 genes have HITS-CLIP peaks and about half of them (4,186) have (T)GCATG-clusters (Fig 3C) [17]. Importantly, 20% of genes with HITS-CLIP peaks, have peaks mapping to the (T)GCATG-clusters (1,687 out of 8,356) (Fig 3D). These data suggest that (T)GCATG-clusters could be significant determinants of Rbfox1 binding to mouse pre-mRNA transcripts.

We next tested whether (T)GCATG-clusters are also present in the human genome by using the same strategy used to scan the mouse genome. Curiously, we found that the human genome has about the same number of (T)GCATG-clusters as found in the mouse genome (19,977 versus 21,005) although 15% more genes in humans contain clusters (9,826 in human versus 8,348 in mouse) (Fig 3E, Fig 3F and S2 Table). Since CLIP data is not available from human brain, we used the eCLIP data obtained from the T-REx-293;RbFox1 human cells containing the TrkB minigene to investigate the relationship between Rbfox1 binding (eCLIP peaks) to (T)GCATG-clusters. This analysis confirmed that the 'canonical' Rbfox1 motif (T)GCATG is the most enriched motif among all the eCLIP peaks and that, as in mouse, the distribution of RbFox1 peaks is mostly present in intronic regions [12,17,21] (S3 Fig). Although HEK293 are non-neuronal cells, we found 4903 genes containing Rbfox1 eCLIP peaks, of which 2,908 (~60%) included (T)GCATG-clusters (Fig 3F). Importantly, 1,237 of the 4903 genes with eCLIP peaks, had eCLIP peaks mapping to (T)GCATG-clusters, a percentage similar to that observed in mouse (25% in HEK293 cells versus 20% in mouse brain) (Fig 3G). Altogether, these data show that Rbfox1 directly binds (T)GCATG-clusters of both human and mouse transcripts.

Moreover, since Rbfox1 is expressed predominantly in brain, heart and skeletal muscles, we analyzed (T)GCATG-clusters presence in all genes associated with diseases in these tissues by exploiting the catalog of human genome-wide association studies (GWAS Catalog). S2 Table lists the genetic coordinates for all the (T)GCATG-clusters mapping on genes associated with muscle disorders, brain and heart diseases found in GWAS.

## The presence of Rbfox1 binding site clusters in human alternatively spliced genes correlates with enhanced activity of Rbfox1 on gene isoform expression

To investigate genome wide whether eCLIP peaks on (T)GCATG-clusters are significant in regulating RbFox1 activity on gene isoform expression we performed RNAseq isoform expression analysis in HEK293-RbFox1 cells in the absence (No Dox) or presence (+Dox) of Rbfox1, as in the eCLIP experiments. Cross-analysis between eCLIP and RNAseq data showed that, among the genes with eCLIP peaks, there is a higher percentage of genes with differential expression of isoforms when eCLIP peaks are present on the (T)GCATG-clusters compared to the group of genes with peaks outside the clusters or with peaks but lacking (T)GCATG-clusters (Fig 3H). Importantly, this difference is maintained irrespective of the change in isoform expression up to a threshold of 100% expression-change (Fig 3H). Moreover, the cumulative average of $\log_2$ fold change in expression was also significantly higher for gene isoforms with eCLIP peaks on clusters ($1.81 \pm 0.063$ $\log_2$FC) compared to those with eCLIP outside of the clusters or with eCLIP peaks but lacking the (T)GCATG-clusters ($1.56 \pm 0.055$ and $1.39 \pm 0.049$ $\log_2$FC respectively; Fig 3I). These data suggest that the presence of (T)GCATG-clusters in the genome is an important determinant of RbFox1 activity on gene isoform expression. To test whether the number of

PLOS Genetics

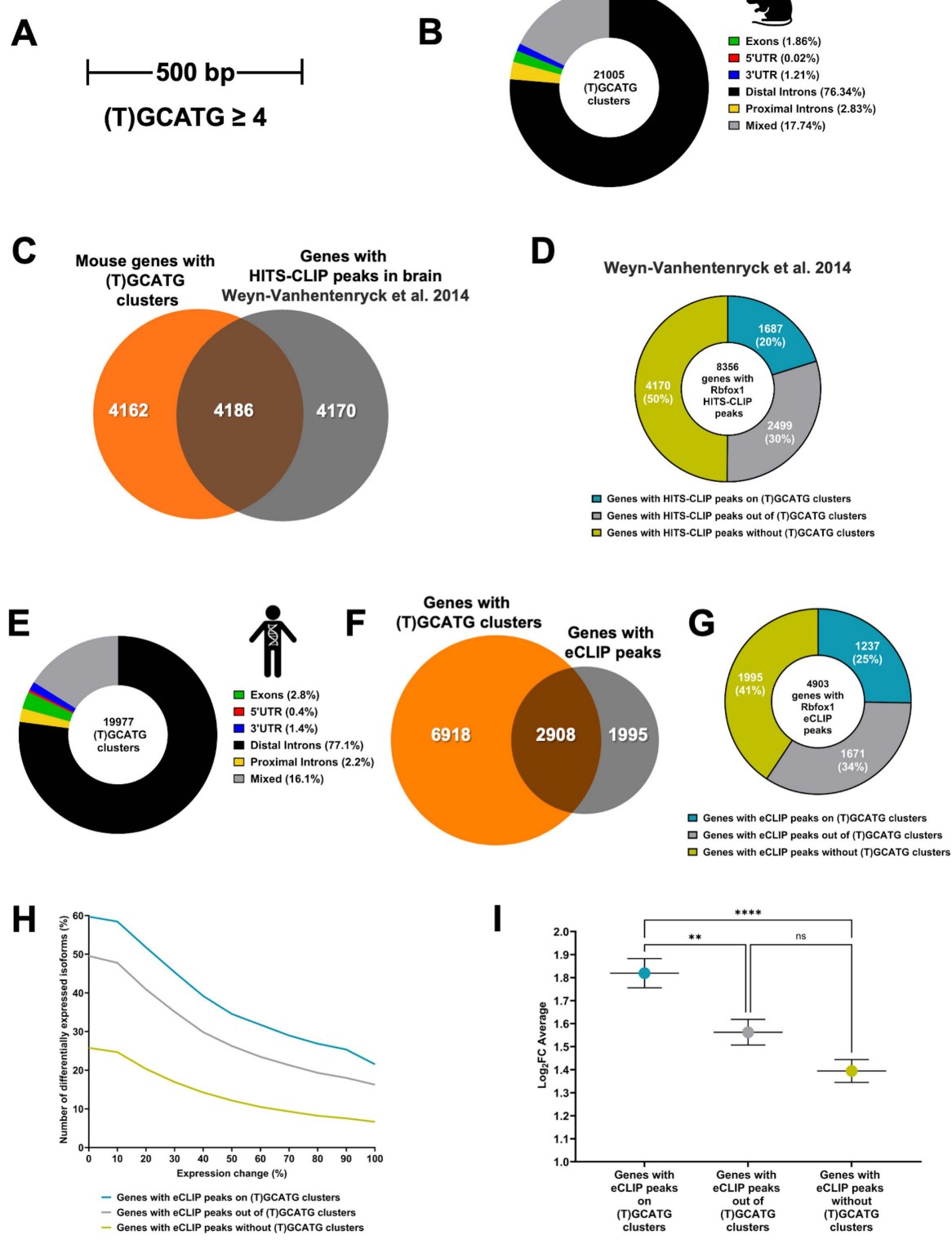

**Fig 3. (U)GCATG-clusters are widespread in the mammalian genome and are associated to RbFox1 isoform gene expression regulation.**
(A) (T)GCATG clusters across the mouse and human genome were defined by the presence of at least four or more motifs within a genomic space of 500 bp. (B) Donut chart showing the distribution in mouse of the 21,005 clusters of motifs between exons, 5'UTR, 3'UTR, proximal introns (within a distance of 500 bp from exons), distal introns and mixed (located in between those categories). (C) Venn diagram showing the overlap between the

total number of genes with (T)GCATG clusters (8348 genes) and the genes with significant Rbfox1 HITS-CLIP peaks (8356 genes) in mouse brain [from [63]]. (D) Donut chart showing the number of genes displaying significant HITS-CLIP peaks on (T)GCATG clusters (1687 genes; 20%) and those without HITS-CLIP peaks on clusters (6669 genes; 80%) among the total number of genes bound by Rbfox1 in mouse brain (8356 genes). (E) Donut chart showing the distribution of the 19,977 (T)GCATG clusters identified in the human genome between exons, 5'UTR, 3'UTR, proximal introns (within a distance of 500 bp from exons), distal introns and mixed (located in between those categories). (F) Venn diagram showing that 2908 genes overlap between the total number of human genes with (T)GCATG clusters (9826 genes) and genes with Rbfox1 eCLIP peaks (4903 genes) identified in the T-REx-293 cell analysis. (G) Donut chart showing the number of human genes displaying significant eCLIP peaks on (T)GCATG clusters (1237 genes; 25%) and the number of genes without eCLIP peaks on (T)GCATG clusters (3,666 total genes; 75%). (H) Chart depicting the relationship between the number of genes (%) with a change in isoform expression and the % change in expression of the isoforms (% Fold Change isoform expression) in the set of genes with and without eCLIP peaks on (T)GCATG clusters. (I) Graph showing the overall average of fold change isoform expression (Log$_2$FC Average) for the set of genes with eCLIP peaks on (T)GCATG clusters and eCLIP peaks outside of the clusters, or eCLIP peaks in the absence of clusters. ** p ≤ 0.01; *** p-value ≤ 0.00001. Image created with permission in BioRender.

RbFox binding sites within a cluster enhances its regulatory effect, we extended our analysis beyond the initial threshold of four binding sites by limiting it to clusters containing more than six and eight (T)GCATG sites. Interestingly, we observed that the average isoform expression change (Log2FC Average) for genes with eCLIP peaks overlapping (T)GCATG clusters remained consistent, regardless of the number of binding sites within the cluster (S4 Fig). Moreover, the number of genes with clusters meeting these higher thresholds decreased sharply as the number of (T)GCATG sites increased: 1237 genes with clusters having ≥4 sites, 279 with ≥6 sites, and only 123 with ≥8 sites. This is why, despite a tendency, the group of genes with clusters with ≥8 sites (123) does not show significance in the average isoform expression change compared to the control group. These findings indicate that increasing the number of RbFox binding sites beyond ≥4 sites within a cluster is not a major determinant of RbFox-mediated splicing regulation.

## Deletion of deep intronic (T)GCATG-clusters eliminates Rbfox1 splicing function on TrkB

To directly test the role of the (T)GCATG-clusters in mediating Rbfox1 function on TrkB isoforms expression we used a recombineering deletion approach in the TrkB-BAC minigene [25]. First, we tested the role of 4 of the 7 eCLIP peaks on RbFox1 regulation of TrkB isoform expression since they were proximal to the TrkB.FL coding sequence (S2A and S2B Fig). Surprisingly, RbFox1 expression in cells with a TrkB-BAC with deletion of the 50 Kb intron immediately upstream of the TrkB.FL coding region still regulated TrkB.FL isoform expression suggesting that the 4 eCLIP peaks in that region were not essential in mediating RbFox1 function (S5 Fig). Next, we targeted the (T)GCATG-clusters by removing, respectively, only about 600 and 400 nucleotides from each of the 50 Kb introns upstream and downstream of the TrkB.T1 exon (S6 Fig). Importantly these short sequences were, respectively, about 25 and 15 Kb upstream and downstream of the TrkB.T1 exon. Removal of each individual cluster independently did not significantly impair Rbfox1 function on TrkB isoform expression when compared to the control BAC (Fig 1C, Figs 4A–4F). Specifically, the TrkB.FL isoform is again strongly upregulated by RbFox1 both at the RNA and protein level in both minigenes with either Cluster 1 or Cluster 2 deleted. The effect on TrkB.T1 levels was less obvious since deleting Cluster 2 did not change its levels compared to the wild-type BAC, while Cluster 1 deletion appeared to render TrkB.T1 levels insensitive to RbFox1 (Figs 4A–4F). Most importantly, the simultaneous deletion of both clusters completely abolished any effect of Rbfox1 (Figs 4G–4I) on the expression levels of both TrkB.FL and TrkB.T1. This result suggest that Rbfox1 regulates gene isoform expression levels by interacting with 'deep' intronic, tightly clustered RNA binding motifs in the immature RNA transcripts.

Lastly, to test whether deletion of the two clusters changed the Rbfox1 binding landscape on the BAC, we performed eCLIP experiments in one of the cell lines harboring the deletion of both Clusters. Interestingly, comparison between the eCLIP peaks of the WT and the mutant minigene showed that only the peaks corresponding to Cluster1 and 2 are missing in the mutant while all other eCLIP peaks were unaltered and unable to compensate for the loss of the missing eCLIP peaks (S7 Fig). These data suggest that removal of the two clusters does not affect RbFox1 binding to the other existing mRNA sites and does not generate new alternative eCLIP peaks.

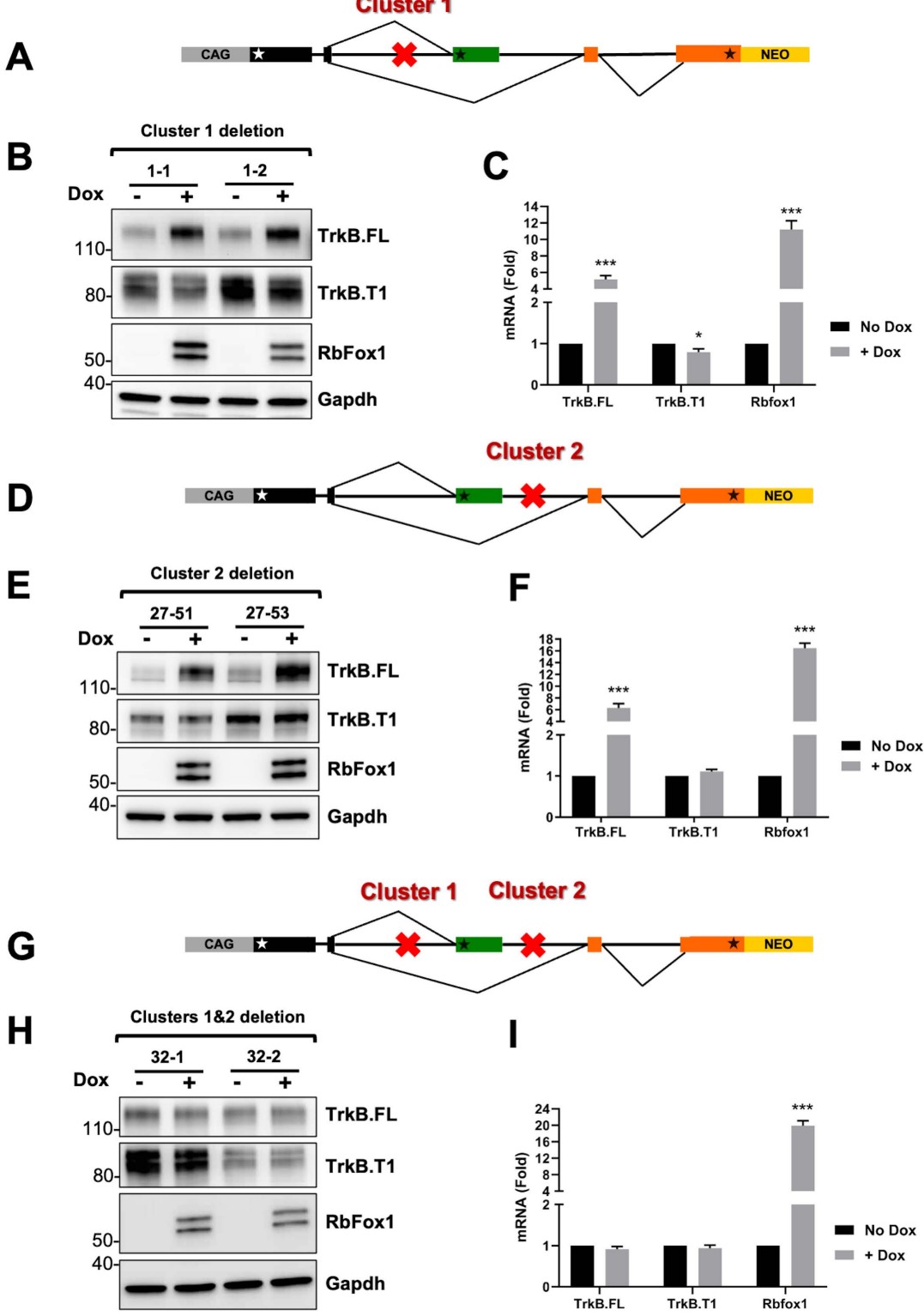

**Fig 4. Deletion of deep intronic (T)GCATG clusters impairs RbFox1 regulation of TrkB isoform expression.** (A) Schematic representation of the location (red X) of the Cluster 1 deletion in the TrkB-BAC minigene analyzed in (B-C). (B) Western blot analysis of TrkB.FL, TrkB.T1 and RbFox1 protein expression levels from two independent cell lines (1−1 and 1−2) with the TrkB-BAC minigene with cluster 1 deleted, in the absence or presence of doxycycline (No Dox or +Dox 0.5 μg/ml for 48h); RbFox1- and Gapdh-specific antibodies were used, respectively, to verify doxycycline induction of

Rbfox1 and as a control of protein loading. (C) Quantitative PCR analysis of *TrkB.FL, TrkB.T1* and *Rbfox1* expression levels in the absence or presence of doxycycline of clones as in (B); n = 6 ± SEM (n = 3 for each clone). (D) Schematic representation of the location of the Cluster 2 deletion in the TrkB-BAC minigene analyzed in (E–F). (E) Western blot analysis as in (B) of two cell lines (27−51 and 27−53) expressing the TrkB-BAC minigene with cluster 2 deleted. (F) Quantitative PCR analysis as in (C) of the two independent cell lines analyzed in (E); n = 6 ± SEM (n = 3 for each clone). (G) Schematic representation of the location of Cluster 1 and 2 deleted in the TrkB-BAC minigene analyzed in (H, I). (H) Western blot analysis as in (B) of the two cell lines (32−1 and 32−2) expressing the TrkB-BAC minigene with cluster 1 and 2 deleted. (I) Quantitative PCR analysis as in (C) of the two independent cell lines analyzed in (H). n = 6 ± SEM (n = 3 for each clone).

## Rbfox2 and Rbfox3 modulate TrkB isoform expression through the (T)GCATG-clusters

RbFox family members including Rbfox1, Rbfox2 and Rbfox3 binds to the same hexanucleotide (U)GCAUG, they participate in the formation of the large assembly of splicing regulators (LASR) complex and, have some but not fully redundant functions [10,20]. Some of the functional differences could be explain by their diversity in the C-terminal domain (CTD), which is a crucial region interacting with the LASR. Thus, to test RbFox family members functional redundancy in regulating splicing through (U)GCAUG clusters, we used the BAC minigene sytem. Expression of Rbfox2 and Rbfox3 in cells with the TrkB-BAC showed a pattern of TrkB isoform regulation similar to that of Rbfox1 although the modulation of TrkB isoforms by RbFox3 appeared significantly reduced compared to RbFox1 and 2 (Fig 5). Importantly, when Rbfox2 or Rbfox3 were expressed in cells carrying the BAC with deletions of cluster 1 and cluster 2, no changes in TrkB isoform expression were observed. These data support the conclusion that the (U)GCAUG clusters are critical for the activity of all Rbfox family members (Fig 5).

## (T)GCATG-clusters are embedded in sequences binding the large assembly of splicing regulators (LASR)

Analysis of the RbFox1 cluster sites in the TrkB minigene showed abundant repetitive elements, including poly-GT sequences (Fig 2C), which have been found to be present in the proximity of CLIP-peaks of Rbfox proteins [20]. Because these sequences are associated with the LASR, a protein complex containing hnRNP M, hnRNP H/F, hnRNP C, Matrin3, hnRNPUL2, NF110/NFAR-2, NF45, and DDX5 [20,32,33], we tested for enrichment of short repetitive sequences within the (T)GCATG-clusters in both mouse and human (Figs 6A–6D respectively). By analyzing all Rbfox clusters with at least four or more (T)GCATG-motifs, we found that motifs for the binding of hnRNP-M (GU-rich pentamers) and hnRNP-H (polyG-rich pentamers), two Rbfox1 interacting partners and members of the LASR, are significantly enriched within the (T)GCATG-clusters when compared to random control sequences with an equal length to the median of the Rbfox-clusters. On the contrary, polyU-rich pentamers binding hnRNP-C, have reduced distribution within the (T)GCATG-clusters as compared to random control sequences (Fig 6A, Fig 6B). A similar analysis was also conducted on 'TG/CA' sequences that are often found in repetitive patterns within introns of genes and can act as splicing regulatory elements [34–36]). By analyzing short tandem repeats (TG/CA)$_6$ including a total of 64 different sequences derived from all the possible (TG/CA)$_6$ combinations we unveiled a significant enrichment of (TG/CA)$_6$ repeats within the Rbfox-clusters, suggesting that clustered (T)GCATG-motifs are embedded in sequences rich in short repeats binding interacting partners involved in splicing regulation (Fig 6C, Fig 6D).

Rbfox1 proteins also interact directly with LASR through their C-terminal domain (CTD). The CTD contains a low-complexity sequence rich in tyrosines which are required for assembly of Rbfox/LASR into higher-order complexes and promote RbFox1 ability to activate splicing [33]. Mutation of the tyrosine residues in the CTD region impairs most of the RbFox1 splicing activity even if it retains the ability of interacting with LASR [33]. We hypothesized that the presence of (T)GCATG-clusters could favor the recruitment of Rbfox1 and the formation of higher-order assembly with LASR by locally increasing the concentration of Rbfox1-proteins. To test this hypothesis, we investigated whether a splicing-defective mutant Rbfox1 (Rbfox1-Y10) containing 10 tyrosine to serine residue mutations in the CTD could retain some splicing activity on the TrkB minigene transcripts [33]. We argued that the presence of (T)GCATG-clusters is

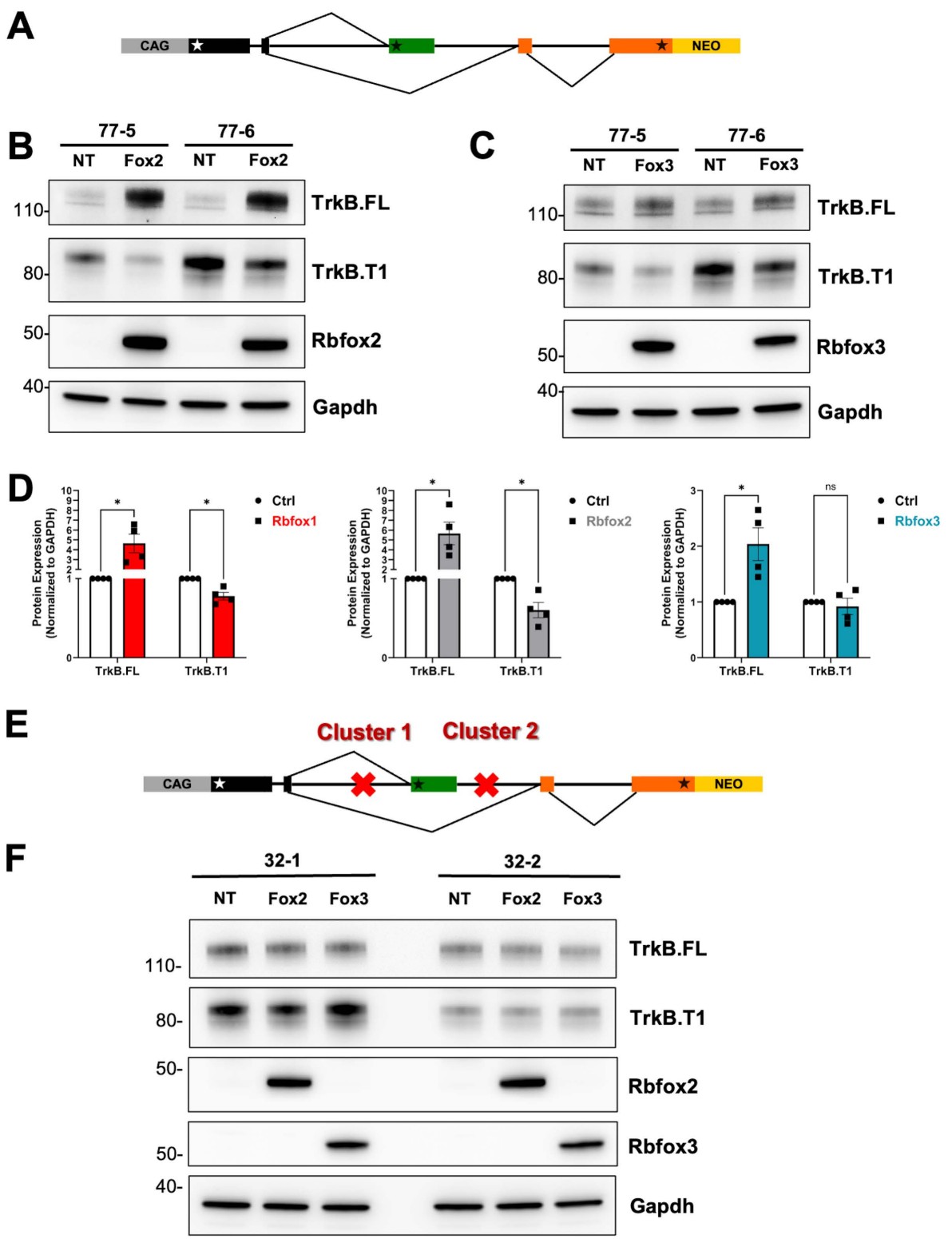

**Fig 5. Rbfox2 and Rbfox3 regulate TrkB isoforms expression through binding to deep intronic (T)GCATG clusters.** (A) Schematic representation of the TrkB-BAC minigene as in Fig 1. (B, C) Western blot analysis of lysates from the two clones expressing the TrkB-BAC minigene (32−1 and 32-2) untransfected (NT) or transfected with plasmids expressing Rbfox2 (Fox2; B) or Rbfox3 (Fox3; C) and probed with an antibody recognizing TrkB.FL or TrkB.T1. Antibodies against RbFox2 and Rbfox3 were used to verify transfection efficiency and Gapdh was used as a loading control. (D) Immunoblot

quantification analysis of TrkB.FL or TrkB.T1 protein levels in HEK293 cells with the WT TrkB-BAC minigene 48h after transfection with Rbfox1 (left panel, as in Fig 1C), RbFox2 (from B, center panel) or Rbfox3 (from C, right panel) relative to not transfected cells (NT); n = 3 ± SEM (One-way ANOVA). (E) Schematic representation of the TrkB-BAC minigene with deletion of Cluster 1 and 2 as in Fig 4G. (F) Western blot analysis as in (B-C) of the two cell lines expressing the TrkB-BAC minigene with cluster 1 and 2 deleted, untransfected (NT) or transfected with plasmids expressing Rbfox2 (Fox2) and Rbfox3 (Fox3).

sufficient to increase the local concentration of RbFox1 proteins in intronic regions to allow, at least partially, the formation of functional higher-order assembly of Rbfox/LASR. Remarkably, we found that the Rbfox1-Y10 mutant retains splicing activity in the TrkB-BAC minigene system, although at reduced levels compared to the wild-type Rbfox1 control (Fig 6E, Fig 6F). As expected, the Rbfox1-Y10 mutant did not have any effect on transcripts splicing generated from the TrkB BAC cluster mutant (S8 Fig). Altogether, these data suggest Rbfox/LASR splicing function also depends on deep intronic clustered (U)GCAUG-motifs which promote local recruitment of multiple RbFox1 proteins to form Rbfox1/LASR higher-order assemblies required for splicing (Fig 7).

## Discussion

Alternative splicing of pre-mRNA is a fundamental genetic process expanding gene function diversity. Several mechanisms regulate the splicing networks during normal differentiation and development including, for example, spatio-temporal expression of specific RBP, regulation of the stoichiometry of different proteins part of the spliceosome, the distribution of RBP motifs present in the pre-mRNA, and RNA sequences that form secondary structures bridging across individual introns to include or skip exons [4,37,38,39]. The size of introns involved in alternative splicing can vary greatly from a few base pairs to hundreds of Kb. Although the biological significance of this variability is unknown it has been speculated that intron and gene size expansion is associated to evolution of increasingly complex gene regulation of the nervous system [40]. However, the elements in deep intronic regions influencing alternative splicing are poorly understood. Here, by studying the role of RbFox1 in regulating alternative splicing of the *Ntrk2* gene we have found that deep intronic, tightly clustered RbFox1 binding motifs are required for the normal expression of Ntrk2 receptor isoforms. Importantly, similar motifs are widespread in the mouse and human genome and appear to mediate RbFox1 regulation of gene expression isoforms in the context of sequences binding the LASR.

Using bacterial genetic engineering technology, we generated an unusually large minigene (~164 Kb) that includes native 50 Kb introns and expresses both major isoforms of the *Ntrk2* gene. This system allowed us to demonstrate that RbFox1 regulates the expression of a TrkB alternatively spliced exon via clusters of RbFox binding sites located as far as 25 Kb upstream and downstream of the exon. Notably, such tightly clustered RbFox binding sites are not unique to the *Ntrk2* gene; they are widespread across both the mouse and human genomes, with more than 75% located in deep intronic regions. The isoform modulation pattern observed in this in vitro system upon RbFox1 expression does not fully replicate what we observed in vivo following RbFox1 overexpression in the hippocampus, where TrkB.T1 isoform levels were upregulated [8]. One reason for the discrepancy could be that the splicing environment likely differs between human HEK293 and murine neurons. Second, one limitation of the system is that the full *TrkB* gene exceeds the size capacity of a single BAC, preventing inclusion of the entire genomic structure. This constraint may affect physiological transcript regulation by omitting many intronic regions normally present in the *TrkB* gene. For example, analysis of the complete *TrkB* sequence revealed another notable region containing 15 clustered (U)GCAUG motifs embedded in highly repetitive sequences in a distal intronic region of the tyrosine kinase domain (S1 Table). It is reasonable to speculate that this (U)GCAUG cluster contributes to in vivo regulation of *TrkB* isoforms, yet it is absent from the TrkB-BAC minigene. Despite these limitations, the TrkB-BAC minigene proved valuable for identifying (U)GCAUG clusters in deep intronic regions and highlights the utility of BAC-based systems for dissecting alternative splicing mechanisms. This approach may offer additional advantages in the study of deep intronic investigations particularly for genes that fall within the BAC size limits.

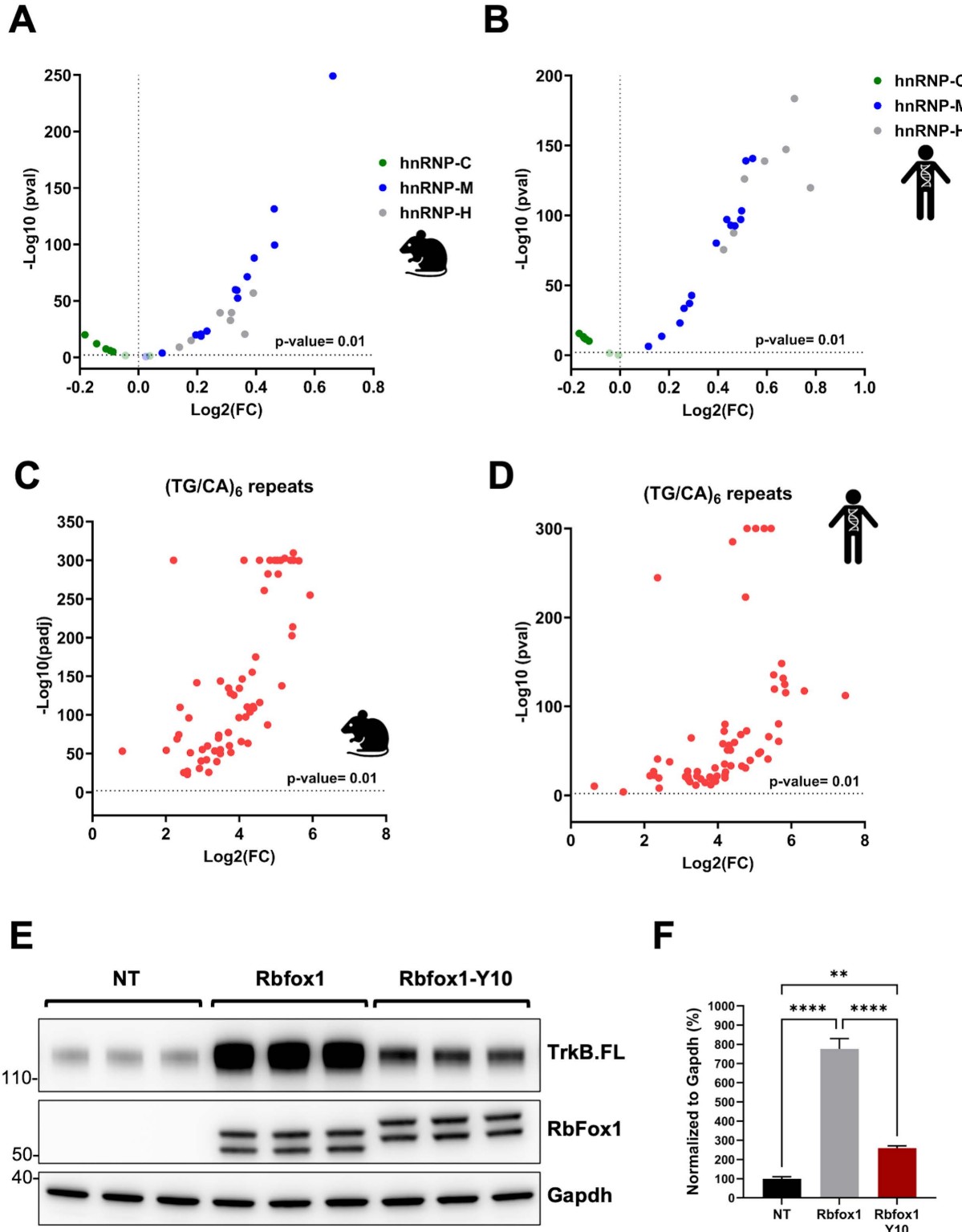

**Fig 6. (T)GCATG clusters are embedded in sequences binding the LASR. (A)** Volcano plot illustrating the enrichment of 14 GU-rich pentamer motifs of hnRNP-M (blue dots) ['TGTTG','GTGTT,'TTGTG','GTTGT','TGTGT','TGGTT','TTGGT','GTGTG','GGTGT','TGTGG','GTGGT','GTTGG', 'TGGTG', 'GGTTG'], seven polyG-rich pentamers motifs of hnRNP-H (gray dots) ['GGGGT', 'GGGGG', 'CGGGG', 'AGGGG', 'TGGGG', 'GGGGC', 'GGGGA'] and seven polyU-rich pentamers of hnRNP-C (green dots) ['ATTTT', 'GTTTT', 'CTTTT', 'TTTTA', 'TTTTC', 'TTTTG', 'TTTTT'] within (T)GCAUG-clusters

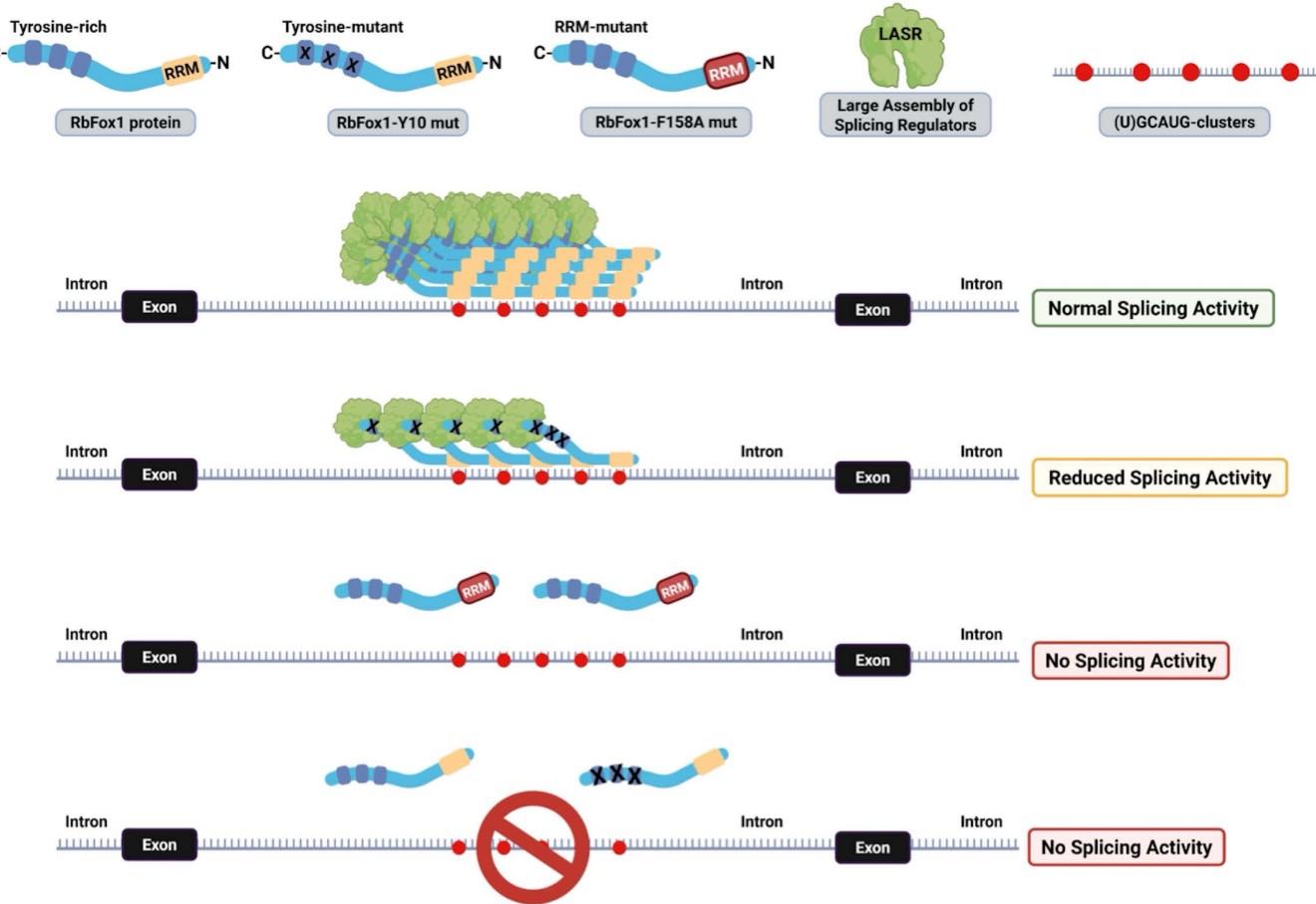

containing at least 4 or more (T)GCATG motifs in the mouse genome. $-Log_{10}$(p-value) is indicated on the Y axis while $Log_{2}$(Fold Change) is indicated on the X axis. P-values greater than 0.01 were considered as statistically significant. **(B)** Volcano plot illustrating the enrichment of motifs of hnRNP-M (blue dots), hnRNP-H (gray dots) and hnRNP-C (green dots) in the human genome analyzed as in (A). **(C)** Volcano plot illustrating the enrichment of $(TG/CA)_{6}$ repeats (64 different combinations) within (T)GCAUG-clusters containing at least 4 or more (T)GCATG motifs in the mouse genome. $-Log_{10}$(p-value) is indicated on the Y axis while $Log_{2}$(Fold Change) is indicated on the X axis. P-values greater than 0.01 were considered as statistically significant. **(D)** Volcano plot illustrating the enrichment of $(TG/CA)_{6}$ repeats (64 different combinations) within (T)GCAUG-clusters in the human genome analyzed as in (C). **(E)** Western blot analysis of lysates from HEK293 cells with the WT TrkB-BAC minigene 48h after transfection with a control (Rbfox1) or an Rbfox1 cDNA with mutations in 10 tyrosine residues in the CTD region [33] showing that the mutant Rbfox1-Y10 partially retains the ability to regulate TrkB isoform expression through (T)GCATG-clusters. Non-transfected cells were used as control (NT). TrkB.FL, Rbfox1 and Gapdh protein levels were analyzed as in Fig 1. **(F)** Immunoblot quantification analysis of TrkB.FL protein levels in HEK293 cells with the WT TrkB-BAC minigene 48h after transfection with Rbfox1 or Rbfox1-Y10 relative to not transfected cells (NT) as in (E); n = 3 ± SEM (One-way ANOVA). Image created with permission in BioRender.

**Fig 7. Model summary of how RbFox1 regulates pre-mRNA splicing in conjunction with LASR through binding of (U)GCAUG-clusters.**
(U)GCAUG-clusters participate in the pre-mRNA splicing regulation by facilitating local recruitment and accumulation of Rbfox1 proteins to pre-mRNA sites where they form large RbFox1 protein aggregates and higher-order assembly with the LASR (normal splicing activity). Conversely, the Rbfox1-Y10 mutant, which has greatly reduced ability to bind LARS has limited splicing efficiency (reduced splicing activity). Inactivating mutations in the RbFox1 RNA-binding motif (RRM) or deleting the (U)GCAUG-clusters leads to a complete loss of RbFox1-mediated splicing function (no splicing activity). C- and N- indicate the carboxy- and amino- terminus, respectively. Image created with permission in BioRender.

Deletion of the downstream cluster of TrkB.T1 (Fig 4D, Fig 4F), while not affecting RbFox1 regulation of the TrkB.T1 isoform, it does influence only the regulation of the TrkB.FL as this isoform level changes following RbFox1 expression. While the reason for this difference is unknown, it is possible that RbFox1 may have separate roles on isoform expression including, regulation of alternative splicing and/or RNA stability [8,12]. To date, a role on RNA stability by RbFox1 has been reported only by its binding to 3'UTR (U)GCAUG site of cytoplasmic RNA [12]. However, our system suggests that (U)GCAUG clusters may mediate this function as well and may be employed to unravel this mechanism.

Large introns are particularly abundant in genes regulating complex and diverse biological functions such as genes expressed in brain, which require a precise spatio-temporal regulation of expression of different isoforms in response to developmental cues as well as environmental stimuli. For example, the neurexins cell adhesion proteins which regulate neuronal development and synaptic function include many different isoforms generated by alternative splicing of genes that are among the largest in the mammalian genome [41,42]. Ntrk3, a member of the Trk family of genes also critical for normal development of the nervous system, generates by alternative splicing two different isoforms, a full-length tyrosine kinase receptor and a shorter isoform [43]. Like TrkB, the alternative spliced exons have, upstream and downstream, two long introns respectively of about 50 and 40 Kb most likely including the elements regulating splicing [44; 45]. Thus, the BAC system provides a unique tool to understand splicing of large introns and the context of the sequences regulating it.

Combining eCLIP data and BAC mutagenesis, we found that deletion of the two clusters surrounding the TrkB.T1 encoding exon, although 50 Kb apart, completely disrupts RbFox1 activity on TrkB isoforms regulation and no changes in eCLIP sites were detected after deletion. This was surprising because there are about 120 additional RbFox binding sites distributed along the introns surrounding the TrkB.T1-encoding exon. However, the clusters with eCLIP peaks are part of regions rich in elements binding other LASR, suggesting that RbFox binding sites are necessary but not sufficient for splicing [33]. Importantly, 4–6 (U)GCAUG-motifs are sufficient to generate functional clusters as clusters with higher numbers of RbFox1 binding sites, are rare and do not appear to have increased functional activity on alternative splicing. In addition, we found that all RbFox family members can regulate alternative splicing of the TrkB-BAC system through binding to distal intronic clusters of (U)GCAUG-motifs but they also displayed a difference in the modulation intensity. This finding suggests that the formation of diverse Rbfox/LASR complexes in these deep intronic areas play an important role in defining how effective these regions are in regulating splicing.

In all, our data support a model in which clustered (U)GCAUG-motifs may be important to initiate the formation of high density RbFox/LASR complexes which drive splicing. Indeed, mutation of the tyrosines in the CTD domain, which allows for RbFox1 physical aggregation and formation of higher-order assembly of Rbfox/LASR only partially abrogate the splicing phenotype in the TrkB-BAC [20,32]. Moreover, the data further supports the notion that the context of the RbFox clusters is critical in determining RbFox1 activity. The finding that removal of only one cluster either upstream or downstream does not cause a significant phenotype, was somewhat puzzling and we cannot exclude that other Rbfox1 binding sites discovered via eCLIP might be participating in this process. However, the fact that the simultaneous deletion of cluster 1 and cluster 2 completely blocks splicing modulation suggests that the remaining five Rbfox-binding regions per se are insufficient to support the Rbfox-dependent modulation, while the presence of the deep intronic (U)GCAUG-clusters are necessary. Intronic sequences responsible for RNA looping could be important determinants in bringing the RbFox1/LASR complex to the appropriate location for splicing [21,46]. Analysis of the 50 Kb sequence between the two clusters may help understand whether RNA bridges that allow for exclusion of the TrkB.T1 exon between the two clusters are possible and may be sufficient to lead to an upregulation of the TrkB.FL isoform [21,47,46]. Furthermore, CRIC-seq global mapping of RNA-RNA interactions mediated by RbFox may help determine how RBFox1 cluster sites are key determinant of alternative splicing [46].

The TrkB-BAC minigene also provides a powerful tool for the functional screening of RbFox1 gene variants isolated from patients with developmental brain abnormalities as it uncovers mutations affecting RbFox1 splicing function [48]. Although its use has so far been limited to the functional testing of RbFox1 variants with mutations within the RNA binding

domain, the finding that it can expose functional defects of RbFox1 with mutations in the tyrosine rich CTD domain suggests a use to test RbFox1 variants with mutations in other domains that may be critical for its interaction with other RBP or proteins of the spliceosome complex.

In summary, we have identified clusters of RbFox1 binding sites embedded in repetitive sequences that can bind the large assembly of splicing regulators. These clusters are important determinants of RbFox1 splicing function and are widely distributed in the mammalian genome, particularly in very deep intronic regions of alternatively spliced genes. RbFox2 and RbFox3 also regulate TrkB isoform splicing, at least in vitro, binding to the same (U)GCAUG sequence, suggesting that this TrkB-BAC minigene could be used to test the functional significance of RbFox protein variants isolated from patients with other pathologies [49]. Moreover, the relevance of RbFox binding clusters in regulating gene isoform expression suggests that these clusters should be added to the list of investigated sequences in the genome for natural mutations in deep intron elements of gene causing brain, heart and skeletal muscle diseases in humans (S2 Table, [48]).

Lastly, our study suggests that BAC-based minigenes up to 200 Kb with unaltered genomic structure could be used to identify and explore, in detail, the function of new elements in deep distal intron regions bound by other RBPs mediating splicing and/or distal interactions [50–53].

## Methods

### TrkB-BAC minigene

The mouse BAC clone RP23-424E11 (Bacpac Genomics) containing 194 kb of the Ntrk2 (TrkB) genomic locus from exon 11 (98 bp; encoding the transmembrane region) to exon 14 (131 bp; corresponding to the second exon of the kinase domain), was modified using the BAC manipulation system as described in [26] to create a functioning minigene that, by alternative splicing, expresses both the truncated (TrkB.T1) and full-length (TrkB.FL) transcript. Briefly, the synthetic CAG promoter and the TrkB extracellular domain cDNA sequence from the 'start' codon (ATG) to the sequence corresponding to mouse exon 10 (36 bp) was placed in frame with exon 11; the cDNA sequence encoding the TrkB kinase domain from the sequence corresponding to exon 15 (173 bp) to the 'stop' codon was placed in frame with exon 14 (Fig 1A), followed by a pGKneo cassette for selection with G418 Sulfate 300 ug/ml (ThermoFisher Scientific 10,131,035) to generate subsequent clonal cell lines. The BAC containing the TrkB minigene was transfected into a T-Rex 293 doxycycline-inducible Rbfox1 expressing cell line as previously described [8] using Targefect per manufacturer instructions (Targeting Systems, Targefect-BAC). Cell ($25x10^3/cm^2$) were induced 24h after seeding with Doxycycline (0.5 mg/ml, D3447, Millipore-Sigma) for RbFox1 expression and harvested 48h later.

Reagents for BAC recombineering technology are available at.
https://frederick.cancer.gov/resources/repositories/Brb/#/recombineeringInformation

### Western blot analysis

Cells were lysed 48h post doxycycline treatment with ice cold RIPA lysis buffer (20–188, Millipore-Sigma) directly in cell culture dishes, scraped, collected and incubated 30 min at 4°C before centrifugation at 13,000 rpm at 4°C. Supernatants were collected and transferred into new tubes. Protein concentrations were quantified using a BCA assay (23,225, ThermoFisher Scientific) and samples were prepared with equal amounts of total protein before adding Laemmli sample buffer 2X (S3401, Sigma-Aldrich). Samples were heated at 95°C for 5 min (protein denaturation step) before loading onto a 4%–12% NuPAGE precast gel for Western analysis (ThermoFisher Scientific). After the transfer to PVDF membranes (LC2005, ThermoFisher Scientific), blots were blocked in 5% non-fat milk in TBS-Tween (0.1%) and incubated overnight at 4°C with the specific antibodies. Primary antibodies were anti-panTrk C15 (used for detecting TrkB.FL - against the intracellular kinase-domain of Trk and recognizing all Trk receptors; sc-139; Santa Cruz) 1:1000 (discontinued; replaced by anti-panTrk ab76291 Abcam, 1:1000), anti-TrkB.T1 C13 (sc-119; Santa Cruz) 1:1000 (discontinued; replaced by

anti-TrkB AF1494 R&D Systems, 1:1000), anti-Rbfox1 (MABE985, clone 1D10, Millipore-Sigma) 1:1000, anti-Rbfox2 (A300-864A, ThermoFisher Scientific) 1:1000; anti-Rbfox3 (MAB377, clone A60, Millipore) 1:1000, anti GAPDH (MAB374; Millipore) 1:5000. After incubation with the appropriate horseradish peroxidase (HRP)-conjugated secondary antibodies (Millipore) 1:5000, membranes were incubated with enhanced chemiluminescent substrate (34,076, ThermoFisher Scientific) for detection of HRP enzyme activity and visualized with a Syngene gel documentation system (GeneSys).

To generate the mutant Rbfox1-Y10, the mouse Rbfox1 cDNA was modified in the C-terminal region converting 10 successive tyrosine residues to serine starting from tyrosine at position 299 and ending with tyrosine 341 as described by Ying and colleagues [34]. WT and mutant Rbfox1-Y10 cDNA were cloned into the mammalian expression vector pcDNA3.1-neo for expression studies.

Rbfox2 and Rbfox3 expression plasmids were obtained from GenScript as pcDNA3.1 expression vectors (Cat.No. OMU19838D, Mouse Rbfox2(NM_001110828) ORF Clone; Cat.No. OMU20110D, Mouse Rbfox3(NM_001039167) ORF Clone).

Cells were transiently transfected 24h after seeding ($25 \times 10^3/cm^2$) using X-tremeGENE 9 DNA Transfection Reagent (6,365,779,001, Millipore-Sigma). Cells were lysed directly in cell culture dishes 48h post-transfection, scraped, collected and screened by Western analysis as described above.

Lysates deglycosylation (S1 Fig) was performed using "Protein Deglycosylation Mix II" (P6044S, New England Biolabs) following the denaturing reaction conditions suggested in the manufacturer's protocol.

## QPCR analysis

Total RNA was extracted from cells using the Qiagen RNeasy Mini kit (Cat.no 74,104) according to the manufacturer's instruction. cDNA was then generated using SuperScript III First-Strand Synthesis System (Cat. No 18,080–051, ThermoFisher Scientific) using oligo-dT and by following the manufacturer's protocol. Real time PCR was performed using BioRad iTaq Universal SYBR-green Supermix (Cat.No. 172–5,120) in a MX3000P (Agilent Technologies) apparatus using the following program: 95°C for 3min; 95°C 10s, 60°C 20s for 40 cycles; 95°C 1min and down to 55°C (gradient of 1°C) for 41 cycles (melting curve step). Delta Ct values were obtained using GAPDH as reference gene.

Student $t$ test was applied for statistical significance assessment.

Primers used:

TrkB common forward: 5′-AGCAATCGGGAGCATCTCT-3′

TrkB.FL reverse: 5′-CTGGCAGAGTCATCGTCGT-3′

TrkB.T1 reverse: 5′-TACCCATCCAGTGGGATCTT-3′

TrkB.T2 reverse: 5′-TCATGAGCCAAAAATGAGTCC-3′

GAPDH forward: 5′-TGCGACTTCAACAGCAACTC-3′

GAPDH reverse: 5′-ATGTAGGCCATGAGGTCCAC-3′

Rbfox1 forward: 5′- TGGCCCCAGTTCACTTGTAT-3′

Rbfox1 reverse: 5′- GCAGCCCTGAAGGTGTTGTA-3′

## Enhanced Cross-Linking Immuno-Precipitation (eCLIP)

eCLIP studies were performed by Eclipse Bioinnovations Inc (San Diego, www.eclipsebio.com) according to the published single-end seCLIP protocol [54] with the following modifications. Approximately 20 million FIp-In-T-Rex-293T cells were UV crosslinked at 400 mJoules/cm2 with 254 nm radiation. The cell pellet was lysed using 1mL of eCLIP

lysis mix and subjected to two rounds of sonication for 4 min, with 30 second ON/OFF at 75% amplitude. 1% of the lysate was treated with Proteinase K to digest the RNA bound proteins. The RNA was then isolated (Zymo) and measured on the TapeStation to assess quality and quantity. The lysate volume equating to 100ug of total RNA was enzymatically digested with a 1:10 dilution of RNase-1 (Ambion) and used as starting material. Validated antibodies were then pre-coupled to Anti-Rabbit IgG Dynabeads (ThermoFisher), added to the digested lysate, and incubated overnight at 4°C. Prior to immunoprecipitation, 2% of the sample was taken as the paired input sample, with the remainder magnetically separated and washed with eCLIP high stringency wash buffers. IP and input samples were cut from the membrane from approximately 75 kDa and above. RNA adapter ligation, IP-Western, reverse transcription, DNA adapter ligation, and PCR amplification were performed as previously described. The eCLIP cDNA adapter contains a sequence of 10 random nucleotides at the 5′ end. This random sequence serves as a unique molecular identifier (UMI) after sequencing primers are ligated to the 3′ end of cDNA molecules [55]. Therefore, eCLIP reads begin with the UMI and, in the first step of the analysis, UMIs were pruned from read sequences using umi_tools (v0.5.1) [56]. UMI sequences were saved by incorporating them into the read names in the FASTQ files to be utilized in subsequent analysis steps. Next, 3′-adapters were trimmed from reads using cutadapt (v2.7) [57], and reads shorter than 18 bp in length were removed. Reads were then mapped to a database of human repetitive elements and rRNA sequences compiled from Dfam [58] and Genbank [59]. All non-repeat mapped reads were mapped to the human genome (hg38) and the custom mouse Ntrk2 minigene using STAR (v2.6.0c) [60]. PCR duplicates were removed using umi_tools (v0.5.1) by utilizing UMI sequences from the read names and mapping positions. Peaks were identified within eCLIP samples using the peak caller CLIPper (https://github.com/YeoLab/clipper) [21]. For each peak, IP versus input fold enrichment values were calculated as a ratio of counts of reads overlapping the peak region in the IP and the input samples (read counts in each sample were normalized against the total number of reads in the sample after PCR duplicate removal). A p-value was calculated for each peak by the Yates' Chi-Square test, or Fisher Exact Test if the observed or expected read number was below 5. Comparison of different sample conditions was evaluated in the same manner as IP versus input enrichment; for each peak called in IP libraries of one sample type we calculated enrichment and p-values relative to normalized counts of reads overlapping these peaks in another sample type. Peaks were annotated using transcript information from GENCODE [61] with the following priority hierarchy to define the final annotation of overlapping features: protein coding transcript (CDS, UTRs, intron), followed by non-coding transcripts (exon, intron).

The eCLIP data generated for this study have been deposited in NCBI's Gene Expression Omnibus and is accessible through GEO Series accession number GSE263173.

### RNA seq analysis

The mRNA-Seq samples were pooled and sequenced using NovaSeq 6,000 S1 Illumina Stranded mRNA Prep and paired-end sequencing files were generated. The quality of the reads was assessed using FastQC [62]. There were 160–179 million reads generated and more than 91% of bases were above the quality score of Q30. Low-quality bases and adapters were trimmed using Cutadapt [63]. Remaining reads were mapped to the human reference genome (hg38) using STAR alignment tool [60]. The average mapping rate of all samples was 88%. Unique alignment was above 80%. Library complexity was measured in terms of unique fragments in the mapped reads using Picard's (Institute, Accessed: 2018/02/21; version 2.17.8 [64]) MarkDuplicate utility. In addition, the gene and isoform-expression estimates were quantified for all samples using RSEM [65]. Differentially expressed transcripts at gene and isoform level were identified using DESEq2 [66] and limma [67] package in R, respectively. The gene-ontology (GO) analysis for differentially expressed genes was carried out using WebGestalt [68]. The RNA-seq data generated for this study have been deposited in NCBI's Gene Expression Omnibus and is accessible through GEO Series accession number GSE263172.

The difference in proportion of differentially expressed isoforms between each condition was assessed using a Chi-Squared test with significance at $P < 0.01$. The difference in expression level of differentially expressed isoforms between each condition was assessed using two-sided Student t *t*est with significance at $P < 0.01$.

### Identification and analysis of (T)GCATG clusters

The UTR exon records were added to the GRCh38 RefSeq annotation file (available on the NCBI Human Genome Resources portal) using the Python script created by David Managadze (https://github.com/yfu/tools/blob/master/add_utrs_to_gff.py). This file was parsed using a custom Python script to find any cluster containing at least 4 (T)GCATG sites in a window of 500 bp. For each identified cluster, additional (T)GCATG sequences less than 500 bp away from the cluster was considered as an extension of the cluster and was added to it. The genomic location, type of location (exon, distal intron, proximal intron, 5'UTR, 3'UTR or mixed), and the number of (T)GCATG sequences were collected for each cluster. The overlap between the genes containing eCLIP peaks and the genes containing clusters was analyzed using a custom R script.

Motifs for the binding of hnRNP-M (14 GU-rich pentamers: 'TGTTG','GTGTT','TTGTG','GTTGT','TGTGT','TGGTT','TTGGT','GTGTG','GGTGT','TGTGG','GTGGT','GTTGG', 'TGGTG', 'GGTTG'), hnRNP-H (seven polyG-rich pentamers: 'GGGGT', 'GGGGG', 'CGGGG', 'AGGGG', 'TGGGG', 'GGGGC', 'GGGGA'), hnRNP-C (seven polyU-rich pentamers: 'ATTTT', 'GTTTT', 'CTTTT', 'TTTTA', 'TTTTC', 'TTTTG', 'TTTTT') and all possible combinations of TG and CA repeats containing at least 6 repeats $(TG/CA)_6$ were counted in the Rbfox-clusters with at least five (T)GCATG-motifs, extended to 50 bp in both directions, using a custom Python script. The same motifs were counted also in 10,000 random sequences with a length equal to the median length of the Rbfox clusters. The frequency of each motif present in the Rbfox-clusters was then compared to the frequency found in the 10,000 random sequences with a length equal to the median length of the Rbfox clusters using a chi-squared test. P-values were FDR-corrected, and plots were generated using ggplot2 package (version 3.4.4) in R.

The human genes associated with brain diseases (9883 genes), muscle disorders (770 genes) and heart diseases (3003 genes) that were generated by the NHGRI-EBI catalog of human genome-wide association studies (GWAS Catalog), were analyzed for the presence of (T)GCATG-clusters and displayed as additional tabs in S2 Table.

## Supporting information

**S1 Fig. (A, B) Western blot analysis of lysates from the two clones with the TrkB-BAC minigene (77−5 and 77−6) before and after deglycosylation and probed with an antibody recognizing TrkB.** FL **(A)** or TrkB.T1 **(B)**. Antibodies against Gapdh were used as a loading control. Note the significant decrease in size of both TrkB.FL and TrkB.T1 bands after deglycosylation and the presence of a single band following deglycosylation. (C, D) RT-PCR analysis of the same clones as in A, B analyzed for truncated TrkB.T1 and TrkB.T2 expression. Note the almost negligeable expression of TrkB.T2 relative to TrkB.T1 further confirming that TrkB.T1 is the main TrkB truncated isoform expressed by the BAC. (TIF)

**S2 Fig. (A, B) eCLIP analysis of RbFox1 in HEK293 cells with the TrkB-BAC as in Fig 2.** eCLIP peaks were derived by subtracting the signal obtained in the absence of RbFox1 (-Dox), considered as background, from the signal from the same cells (line 77−5 from Fig 1) expressing RbFox1. In green are seven statistically significant eCLIP peaks, all in distal intronic regions. Numbers under each eCLIP peak indicate the p-value ($-\log_{10}$). Below are the enlargements of the areas containing the eCLIP peaks (green) not depicted in Fig 2 relative to the position of RbFox1 (T)GCATG binding motifs (red). (C) Location, statistics, and fold change enrichment of the seven eCLIP peaks found across the minigene sequence shown in Fig 2. (TIF)

**S3 Fig. Analysis of Rbfox1-eCLIP peaks in HEK293 cells with induced Rbfox1 expression (+Dox) compared to uninduced cells (No Dox). (A)** Top five enriched motifs identified in CLIP-seq peaks by the HOMER motif analysis. **(B)** Pie chart depicting the relative frequency of eCLIP peaks that map to each specific gene region (with a peak $Log_2$ fold enrichment ≥ 3 and p-value ≤ 0.001). **(C, D)** Peak Metagene Plot, depicting the average number of peaks mapped to the specific genomic regions indicated in B. The number of peaks was calculated for each gene region followed by normalization with the length of the regions. The average number of peaks was then calculated for a set number of positions along the regions.
(TIF)

**S4 Fig. Increasing the number of (T)GCATG motifs in clusters does not increase the gene isoform expression change.** Graph showing the average of fold change isoform expression ($Log_2$FC Average) for the set of genes with eCLIP peaks on clusters with different number of (T)GCATG motifs (≥ 4, 6 or 8) compared to genes with eCLIP peaks without (T)GCATG clusters. Note the dramatic inverse correlation in the number of genes with eCLIP peaks as clusters have increased (T)GCATG motifs number. ** p ≤ 0.01; *** p-value≤ 0.00001; ns, non-significant.
(TIF)

**S5 Fig. Removal of an intron containing 4 eCLIP peaks in the TrkB-BAC does not influence RbFox1 ability to regulate TrkB splicing. (A)** Schematic representation of the WT TrkB-BAC and the TrkB-BACs with deletion of the intron immediately upstream of the TrkB kinase coding region. **(B)** Western blot analysis of TrkB.FL and TrkB.T1 protein expression levels from two independent cell lines (A2 and A4) with the TrkB-BAC minigene with the intron deletion, in the absence or presence of doxycycline (No Dox or +Dox 0.5 µg/ml for 48h); RbFox1- and Gapdh-specific antibodies were used, respectively, to verify doxycycline induction of Rbfox1 and as a control of protein loading.
(TIF)

**S6 Fig. Strategy to delete (T)GCATG-clusters from the TrkB-BAC minigene. (A)** Schematic representation of the TrkB-BAC minigene showing the position of the PCR-primers designed to detect the deletion of (T)GCATG-cluster 1 and (T)GCATG-cluster 2 (indicated by red X) analyzed in (C). **(B)** magnification of Cluster 1 and 2 areas indicating the location and sequence of the primers used for the analysis. **(C)** PCR analysis of genomic DNA from HEK293 cells used as negative control, cell lines expressing the 'wild-type' TrkB-BAC minigene (77−5 and 77−6 cells), cell lines expressing the TrkB-BAC minigene with cluster 1 deletion (1−1 and 1−2 cells), cell lines expressing the TrkB-BAC minigene with cluster 2 deletion (27−51 and 27−53 cells) and cell lines expressing the TrkB-BAC minigene with both cluster 1 and 2 deletion (32−1 and 32−2 cells). The PCR detecting cluster 1 deletion shows an amplicon of 891 bp (wild-type minigene sequence) and an amplicon of 279 bp (deletion of cluster 1). The PCR detecting cluster 2 deletion shows an amplicon of 746 bp (wild-type minigene sequence) and an amplicon of 322 bp (deletion of cluster 2).
(TIF)

**S7 Fig. Deletion of RbFox1 binding sites in Cluster 1 and 2 leads to the loss of only Cluster 1 and 2-specific eCLIP peaks. (A)** Schematic representation of the TrkB-BAC minigene indicating the location of cluster1 and 2 deletion (red x) with the RbFox1 (+ Dox) eCLIP analysis of wild type TrkB-BAC expressing cells (line 77−5) compared to TrkB-BAC cells with cluster 1 and 2 deletions (line 32−1). Note the presence of only two statistically significant eCLIP areas in the intronic regions (black marks) corresponding to cluster 1 and 2. **(B)** Enlargement of the areas containing the two eCLIP peaks (indicated by shadowed arrows) corresponding to the (T)GCATG clusters (red marks). **(C)** Location, statistics, and fold change enrichment of all eCLIP peaks located in cluster 1 and cluster 2 in (B).
(TIF)

**S8 Fig. Mutant Rbfox1-Y10 does not regulate TrkB isoform expression in the TrkB-BAC minigene lacking Rbfox1 binding cluster 1 and 2. (A)** Schematic representation of the TrkB-BAC minigene indicating the location of cluster1 and 2 deletion (red x). **(B)** Western blot analysis of lysates from HEK293 cells with the mutant cluster 1 and 2 deletion TrkB-BAC minigene 48h after transfection with a control (RbFox1) or an RbFox1 cDNA with mutations in 10 tyrosine residues in the CTD region [66] (Rbfox1-Y10). Non-transfected cells were used as control (NT). TrkB.FL, Rbfox1 and Gapdh protein levels were analyzed as in Fig 1.
(TIF)

**S1 Table. List and genomic coordinates of mouse (T)GCATG-clusters.**
(XLSX)

**S2 Table. List and genomic coordinates of human (T)GCATG-clusters, including tabs of subgroups of GWAS genes with (T)GCATG-clusters associated with skeletal muscle, heart and brain diseases**
(XLSX)

## Acknowledgments

We would like to thank Eileen Southon and Jodi Becker for critical reading of the manuscript, Thomas Gonatopoulos and Zhi-Ming Zheng for suggestions and input on the study, Bao Tran, Yongmei Zhao and Jyoti Shetty of the Sequencing Facility of the Frederick National Laboratory for Cancer Research for their help with the RNAseq experiments, Kylie Shen and Heather Foster from Eclipse Bioinnovations for the eCLIP analysis. This research was supported by the Intramural Research Program of the National Institutes of Health (NIH). The contributions of the NIH authors were made as part of their official duties as NIH federal employees, are in compliance with agency policy requirements, and are considered Works of the United States Government. However, the findings and conclusions presented in this paper are those of the authors and do not necessarily reflect the views of the NIH or the U.S. Department of Health and Human Services.

## Author contributions

**Conceptualization:** Francesco Tomassoni-Ardori, Lino Tessarollo.

**Data curation:** Francesco Tomassoni-Ardori, Melissa Galloux, Lino Tessarollo.

**Formal analysis:** Francesco Tomassoni-Ardori, Melissa Galloux.

**Investigation:** Francesco Tomassoni-Ardori, Mary Ellen Palko.

**Methodology:** Francesco Tomassoni-Ardori, Melissa Galloux, Lino Tessarollo.

**Resources:** Lino Tessarollo.

**Software:** Melissa Galloux.

**Supervision:** Lino Tessarollo.

**Validation:** Mary Ellen Palko, Lino Tessarollo.

**Visualization:** Lino Tessarollo.

**Writing – original draft:** Francesco Tomassoni-Ardori, Mary Ellen Palko, Lino Tessarollo.

**Writing – review & editing:** Francesco Tomassoni-Ardori, Lino Tessarollo.

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
