## [Decision Letter · Decision Letter 0]

23 Apr 2025

PGENETICS-D-25-00335

Clusters of deep intronic RbFox motifs embedded in large assembly of splicing regulators sequences regulate alternative splicing

PLOS Genetics

Dear Dr. Tessarollo,

Thank you for submitting your manuscript to PLOS Genetics. After careful consideration, we feel that it has merit but does not fully meet PLOS Genetics's publication criteria as it currently stands. Therefore, we invite you to submit a revised version of the manuscript that addresses the points raised during the review process.

Please submit your revised manuscript within 60 days Jun 22 2025 11:59PM. If you will need more time than this to complete your revisions, please reply to this message or contact the journal office at plosgenetics@plos.org. Please include the following items when submitting your revised manuscript:

We look forward to receiving your revised manuscript.

Kind regards,

Edward Chuong

Academic Editor

PLOS Genetics

Monica Colaiácovo

Section Editor

PLOS Genetics

Aimée Dudley

Editor-in-Chief

PLOS Genetics

Anne Goriely

Editor-in-Chief

PLOS Genetics

**Additional Editor Comments :**

3 reviewers have assessed your manuscript, and are overall supportive of the topic and the study on deep intronic splicing regulatory elements. Each raise multiple concerns about experimental/analysis details and interpretation that would need to be addressed prior to publication.

**Journal Requirements:**

At this stage, the following Authors/Authors require contributions: Francesco Tomassoni-Ardori, Mary Ellen Palko, Melissa Galloux, and Lino Tessarollo. Please ensure that the full contributions of each author are acknowledged in the "Add/Edit/Remove Authors" section of our submission form.

The list of CRediT author contributions may be found here: https://journals.plos.org/plosgenetics/s/authorship#loc-author-contributions

- ® on page: 23.

4) We have noticed that you have uploaded Supporting Information files, but you have not included a complete list of legends. Please add a full list of legends for your Supporting Information files (Supplementary Tables) after the references list.

Potential Copyright Issues:

i) Figures 3B, 3E, 5, and 6. Please confirm whether you drew the images / clip-art within the figure panels by hand. If you did not draw the images, please provide (a) a link to the source of the images or icons and their license / terms of use; or (b) written permission from the copyright holder to publish the images or icons under our CC BY 4.0 license. Alternatively, you may replace the images with open source alternatives. See these open source resources you may use to replace images / clip-art:

2) State what role the funders took in the study. If the funders had no role in your study, please state: "The funders had no role in study design, data collection and analysis, decision to publish, or preparation of the manuscript.".

**Reviewers' comments:**

Reviewer's Responses to Questions

Reviewer #1: The manuscript by Tomassoni-Ardori et al. presents the discovery of two clusters of (U)GCAUG motifs flanking the TrkB.T1’s coding exons. These motifs are embedded within sequences bound by a large assembly of splicing regulators. Through bioinformatic analysis of multiple CLIP-seq and RNA-seq datasets, the authors propose that these deeply embedded clusters of the RbFox1-binding motif (U)GCAUG might be a crucial element in regulating the alternative splicing of numerous genes in the brain. They further conducted experiments and found that the deletion of these two clusters abolishes RbFox1's splicing regulatory function on TrkB, and the RbFox1-Y10 mutant still exhibits relatively lower splicing regulatory activity. This manuscript contributes to a better understanding of RbFox1's role in regulating TrkB expression. Overall, the findings are interesting; however, several issues need to be addressed.

1. In Figs. 1 and 4, particularly in Fig. 4, how can the authors explain the multiple bands of western signals for both the FL and T1 proteins? Are these bands a result of other alternative splicing events occurring in the related region? To clarify this, RT-PCR results from the corresponding samples should be provided to reveal the details of alternative splicing in each assay.

2. In Fig. 2, how were the boundaries of the two clusters determined? Deleting such long regions could directly impact the binding of other splicing factors, as the authors mentioned regarding the LASR complex. Testing mutations of the (U)GCAUG motifs would be beneficial in addressing this concern.

3. What about the other five RbFox1-binding regions flanking the TrkB.T1-coding exons? If these regions were deleted, would similar effects on TrkB alternative splicing be expected? Conducting such tests could confirm the direct role of these clusters, independent of the effects of Rbfox1 abundance.

4. In Figs. 4B and 4E, when one of the clusters is deleted, the expression of TrkB.FL increases. However, the level of TrkB.T1 either remains unchanged or even increases. How can this be explained?

5. In Fig. 3C, the numbers 8,348 and 8,356 both incorporate the middle value 4,186. It would be more straightforward to subtract 4,186 from 8,348 and 8,356, respectively. Otherwise, it may confuse. The same problem exists in Fig. 3F.

6. In line 187, the text reads "at least four or more (T)GCATG elements," while in line 273, it says "at least five or more (T)GCATG-motifs." Why are there two different criteria?

Reviewer #2: In their paper titled “Clusters of deep intronic RbFox motifs embedded in large assembly of splicing regulators sequences regulate alternative splicing,” Tomassoni-Ardori et al. identify clusters of Rbfox1 sites in large introns of TrkB that regulate the splicing of the TrkB.FL and TrkB.T1 isoforms. They construct a BAC minigene, and demonstrate that these sites are directly bound by Rbfox1, contain elements suggesting they are bound by LASR complex components, and when deleted alter the splicing pattern of TrkB. The authors further show that thousands of genes contain clusters of Rbfox1 sites in distal introns, and using available HITS-CLIP data show these clusters are indeed direct binding targets, and genes with bound Rbfox1 site clusters have higher levels of isoform expression diversity. The clusters are enriched in LASR-associated sequences, and the Rbfox1-Y10 mutant, previously shown to impair splicing activity and LASR formation, reduces levels of reporter expression.

The data on TrkB regulation by Rbfox1 is straight-forward and convincing. The BAC minigene is an elegant tool, and has a strong potential for use in future studies to understand Rbfox1 and LASR function. The paper is an important contribution to the field, and will be of interest to the splicing and Rbfox1 fields. The data on CTD function and LASR involvement is less convincing, and the text should be modified to reflect alternate interpretations of these results. I have the following suggestions to improve the manuscript:

The number of genes with peaks in Figure 3C and discussed in the text (Lines 190-Line 193) do not agree. The authors should use the actual counts (8348 and 8356) rather than “about 8300”. In addition, this Venn Diagram seems to be incorrectly labeled, as it would suggest that there are 8356 + 4186 or 12542 genes with HITS-CLIP peaks in the Weyn-Vanhentenryck data and 8348 + 4186 or 12534 genes in the mouse genome. The Venn Diagram in Figure 3F seems to have the same issue. The total number of genes in the circle should be moved outside of the circle, or it looks like it denotes the number of genes in a specific region of the circle.

Can the authors comment on the number of Rbfox1 binding sites required in a “cluster” region to retain the effects they report here? They identify clusters as more than 5 (T)GCATG sites, but what is the range of binding sites in these regions? Are clusters with for example >10 sites common and more effective? Within the vicinity of the other eCLIP peaks in TrkB, how many (T)GCATG sites are present? Are the clustered Rbfox1 sites the key element to these clusters, or would they still be function with 1-2 Rbfox1 sites? In other words, are Rbfox1 site clusters the defining feature of these elements, or do the concentration of other LASR binding elements play a role?

The authors overstate the outcome of their experiments with Rbfox1-Y10. The authors infer details on the assembly of LASR based on use of Rbfox1-Y10, but they do not test assembly of LASR directly. Rbfox1-Y10 splicing levels of TrkB are indeed significantly reduced in Figure 5E,F. In lines 303-304 the authors claim the data suggest the CTD requirement to mediate higher-order assembly of LASR for splicing activation by Rbfox is not absolute. This interpretation may not be correct, as Ying et al., 2017 show that the Y10 mutant can mediate some splice events (see in particular Figure 5A and 4D in that paper) and that region C3 of the CTD can interact (albeit weakly) with LASR. It is therefore not clear that the Rbfox1-Y10 should abrogate interaction with LASR or completely inhibit splicing. The authors’ data is consistent with a strongly reduced function of Rbfox1, but the CTD likely still has some functionality with Rbfox1-Y10.

The authors need to add citations to relevant regions of the discussion, for example lines 312-315 and 333-336. These concepts seem to be from the literature, but citations are missing. In addition, the authors could expand their discussion of RNA-looping, which is a very intriguing point suggested by their results, but they only cite a single paper (Lovci et al, 2013). In addition, the authors could provide another example or two of long-intron containing genes where the long introns are critical to splicing regulation, as the only long-intron containing example provided is TrkB.T1 (lines 334-338), which is the topic of the paper.

The Methods section needs additional details. Line 403. In the methods, the length of Dox treatment before harvest should be included. What confluency were the cells grow to before Dox treatment? The confluency and length of transfection also should be included in line 428. Line 407. Were the cells lysed directly in the cell culture dish? Lines 416-420. What concentrations were the primary and secondary antibodies used at? Line 433. What cDNA generated using oligo-dT or random primers? Were there any modifications to the SSIII protocol?

Reviewer #3: The authors address the function of deep intron splicing regulatory elements, an under-studied phenomenon that likely plays a critical role in splicing of many genes. They use BAC-based minigenes, an interesting approach to identify and explore the function of new elements in deep distal intron regions. The TrkB gene was used as the model for study,

The data clearly show that deep intron clusters of Rbfox binding sites flanking the terminal T1 exon strongly influence the balance of T1 vs FL isoforms in RNA expressed from the BAC minigene. Moreover, an Rbfox1 mutation that interferes with binding was able to abrogate this effect on splicing. These results certainly support the general hypothesis that deep intron sequences can modulate splicing of exons mapping tens of kb distant from the regulatory motifs.

However, there is concern about whether the specific BAC minigene used in this study accurately represents the in vivo situation. One might expect the first experiment would be to test whether the minigene behaves like the endogenous gene with respect to splicing regulation by Rbfox1. But the authors’ 2019 paper in Elife reported that “Rbfox1 upregulation increases TrkB.T1 expression in vivo”, relative to TrkB.FL. In contrast, the current manuscript observes the converse, an increase in TrkB.FL isoform expression associated with over-expression of Rbfox1. Is there a logical explanation for the disparity that would justify the use of this model?

1. The authors’ bioinformatic analyses demonstrate clearly that (U)GCAUG clusters are widely present in both mouse and human genes. Further, they show that genes with deep intron (T)GCATG clusters mapping to Rbfox1-eCLIP sites are more likely to be alternatively spliced in an Rbfox1-dependent manner than genes lacking this Rbfox1 cluster/eCLIP overlap. These results certainly support the main hypothesis that deep intron (T)GCATG clusters are important regulators of alternative splicing.

2. The authors also show that these (T)GCATG clusters are enriched for motifs known to bind other components of the LASR complex, supporting the model in which Rbfox1 functions in the context of LASR complexes. However, the authors may want to temper the conclusion that “clustered (U)GCAUG-motifs promote the recruitment of RbFox1 proteins to form a Rbfox1/LASR complex required for splicing” (from the abstract). While this is a reasonable hypothesis, they haven’t actually demonstrated the presence of LASR complexes.

3. Line 219: More information is needed to evaluate the significance of the statement that “we found that about 40% of the genes involved in these pathologies have (T)GCATG-clusters including brain disease 36.5%, muscle disorders 40.8% and heart disease 39.5%” ? Are these percentages higher than the percent of all expressed genes in these tissues having (T)GCATG clusters? If 9826 human genes have (T)GCATG clusters in all, is this close to 40% of all genes?

4. Figure 3C, F. Other reviewers can correct me if I’m wrong, but I think the numbers in the Venn diagrams are not presented in the conventional manner. I think the labels outside of the shared regions should represent the numbers of unshared events, not the total events in the encompassing circle.

5. Figure 3H. I don’t understand the figure due to some confusion about the axes. The X-axis is labeled as fold change; what does 100% fold change mean? I think a two-fold change in isoform expression would be a 100% increase, correct? Are there no examples of a greater than two-fold change in isoform expression? Can I interpret the graph to mean that ~25% of genes with (T)GCATG clusters on eCLIP peaks show a 100% (2-fold) change in isoform expression? If I’m interpreting this correctly, a change in isoform expression from 0.5% to 4% would be more significant (higher fold change) than a change from 20% to 80%, but one would think the latter might be more biologically significant.

6. Figure 6. Since proteins are generally presented with N-termini at the left and C-termini at the right, the diagrams in Figure 6 seem backwards (the tyrosine should be C-terminal relative to the RRM). This could be clarified by labeling the N- and C-termini.

7. The 2023 paper by Li et al., in JCI, does not have the complete citation.

**Have all data underlying the figures and results presented in the manuscript been provided?**

Reviewer #1: Yes

Reviewer #2: Yes

Reviewer #3: Yes

PLOS authors have the option to publish the peer review history of their article (what does this mean? ). If published, this will include your full peer review and any attached files.

**Do you want your identity to be public for this peer review?** For information about this choice, including consent withdrawal, please see our Privacy Policy .

Reviewer #1: No

Reviewer #2: No

Reviewer #3: No

**Figure resubmission:**
---

## [Decision Letter · Decision Letter 1]

26 Aug 2025

Dear Dr Tessarollo,

We are pleased to inform you that your manuscript entitled "Clusters of deep intronic RbFox motifs embedded in large assembly of splicing regulators sequences regulate alternative splicing" has been editorially accepted for publication in PLOS Genetics. Congratulations!

Yours sincerely,

Edward Chuong

Academic Editor

PLOS Genetics

Monica Colaiácovo

Section Editor

PLOS Genetics

Aimée Dudley

Editor-in-Chief

PLOS Genetics

Anne Goriely

Editor-in-Chief

PLOS Genetics

Comments from the reviewers (if applicable):

Reviewer's Responses to Questions

**Comments to the Authors:**

Reviewer #1: The authors have addressed my concerns and made corresponding revisions. I have reviewed the modified manuscript. At this time, I have no further questions.

Reviewer #2: The authors provide a significantly revised manuscript that addresses all of my previous concerns. They have revised portions of the text, improving clarity. They added references as requested to sections in the discussion. They have also added multiple new experiments, including identifying potential redundancy in function of RbFox1 with RbFox2 and RbFox3.

While the BAC may not fully replicate endogenous regulation, it provides a valuable system to identify regulatory principles of Rbfox1 and LASR function. The ability to test these principles in a system where both the target as well as the levels of the primary regulator (Rbfox1) can be manipulated provides an elegant means to dissect regulatory function.

Reviewer #3: The authors have answered the previous critiques very well. I agree with their main conclusions that the data support regulation of alternative splicing by deep intron clusters of RBFOX binding motfs, based on experiments with BAC constructs and RBFOX expression clones. I am less convinced that the system accurately reports on the specific regulation of NTrk2 splicing. but the authors have appropriately tempered that discussion in the revised manuscript.

**Have all data underlying the figures and results presented in the manuscript been provided?**

Reviewer #1: Yes

Reviewer #2: Yes

Reviewer #3: Yes

PLOS authors have the option to publish the peer review history of their article (what does this mean? ). If published, this will include your full peer review and any attached files.

**Do you want your identity to be public for this peer review?** For information about this choice, including consent withdrawal, please see our Privacy Policy .

Reviewer #1: No

Reviewer #2: No

Reviewer #3: No

**Data Deposition**

http://datadryad.org/submit?journalID=pgenetics&manu=PGENETICS-D-25-00335R1

**Press Queries**

---

## [Editor Report · Acceptance letter]

PGENETICS-D-25-00335R1

Clusters of deep intronic RbFox motifs embedded in large assembly of splicing regulators sequences regulate alternative splicing

Dear Dr Tessarollo,

We are pleased to inform you that your manuscript entitled "Clusters of deep intronic RbFox motifs embedded in large assembly of splicing regulators sequences regulate alternative splicing" has been formally accepted for publication in PLOS Genetics! Your manuscript is now with our production department and you will be notified of the publication date in due course.

With kind regards,

Anita Estes

PLOS Genetics

On behalf of:
